# Instabilities, Dynamics, and Energetics accompanying Atmospheric Layering (IDEAL): High-Resolution in-situ Observations and Modeling in and above the Nocturnal Boundary Layer

Abhiram Doddi[1], Dale Lawrence[1], David Fritts[2], Ling Wang[2], Thomas Lund[2], William Brown[3], Dragan Zajic[4], and Lakshmi Kantha[1]

[1]Smead Aerospace Engineering Sciences, University of Colorado, Boulder, CO, USA
[2]GATS, Boulder, CO, USA
[3]Earth Observing Laboratory, National Center for Atmospheric Research, Boulder, CO, USA
[4]Meteorology Division, Dugway Proving Ground, Dugway, UT, USA

**Correspondence:** Abhiram Doddi (abhiram.doddi@colorado.edu); Dale Lawrence (dale.lawrence@colorado.edu); David Fritts (dave@gats-inc.com); Ling Wang (lwang@gats-inc.com); Thomas Lund (t.lund@gats-inc.com); William Brown (wbrown@ucar.edu); Dragan Zajic (dragan.zajic.civ@mail.mil); Lakshmi Kantha (kantha@colorado.edu)

**Abstract.** The Instabilities, Dynamics, and Energetics accompanying Atmospheric Layering (IDEAL) program was conceived to improve understanding of the dynamics of thin strongly stratified "sheet" and deeper, weakly stratified "layer" (S&L) structures in the lower troposphere under strongly stable conditions. The field portion of the IDEAL program was conducted from 24 October to 15 November 2017 at Dugway Proving Ground, Utah to target nighttime lower troposphere S&L conditions. It employed a synergistic combination of observations by multiple, simultaneous DataHawk-2 (DH2) small unmanned aircraft systems (sUAS) and concurrent ground-based profiling by an NCAR Earth Observing Laboratory Integrated Sounding System (ISS) comprising a wind profiler radar and hourly, high-resolution radiosonde soundings. DH2 measurement intervals, vertical (~2-4 km) and horizontal (~5-10 km) flight trajectories were chosen based on local high-resolution weather forecasting, and guided by near-real-time ISS measurements. These flights combined simultaneous vertical and slant-path profiling, and/or horizontal racetrack sampling, spanning several hours before sunrise. High spatial and temporal resolution data were down-linked real time to enable near-real-time changes in DH2 flight paths based on observed flow features. The IDEAL field program performed 70 DH2 flights on 16 days, coordinated with 93 high-resolution radiosonde soundings. Raw and derived measurements from this campaign are outlined, and preliminary analyses are briefly described. This data set, as well as "quick look" figures are available for access by other researchers, as described in the paper.

## 1 Introduction

Under stable conditions, the vertical structure of the atmosphere is characterized by thin, strongly stable non-turbulent "sheets" separated by thicker, less stable and often weakly turbulent "layers" (Woods, 1969, 1968; Gage and Green, 1978; Röttger and Liu, 1978). These sheet and layer (S&L) structures are often observed in temperature, humidity, and horizontal winds within the lower troposphere (Balsley et al., 2003, 2006; Chimonas, 1999; Mahrt, 1999; Xing-Sheng et al., 1983; Kantha

et al., 2019) and into the edge of the Stratosphere (Barat, 1982; Fairall et al., 1991; Gage and Balsley, 1980; Röttger, 1980; Woodman and Guillen, 1974). The S&L structures are known to play an important role in the transport and mixing of heat, momentum, and constituents (Barat, 1982; Chimonas, 1999; Dalaudier et al., 1994; Hunt et al., 1985), as well as important roles in optical (Coulman et al., 1995) and radio (Gossard et al., 1984; Luce et al., 2001; Röttger, 1980; Xing-Sheng et al., 1983) wave propagation.

The large-scale vertical features of the layering structures have been qualitatively analyzed using monostatic and bistatic VHF radar observations (Balsley et al., 2006, 2003; Dalaudier et al., 1994; Luce et al., 2001, 1995; Woodman and Chu, 1989). Small-scale details have been characterized in terms of typical sheet thickness and stability, thickness of turbulent layers, Richardson number, and turbulence Reynolds number through in-situ measurements from radiosonde soundings, stationary observation towers and tethered lifting systems (Balsley et al., 2003, 2006; Muschinski et al., 2001a), and more recently, using aircraft (Lawrence and Balsley, 2013; Muschinski and Wode, 1998; Scipión et al., 2016). High-resolution multipoint measurements of temperature (Barat, 1982; Coulman, 1973; Frehlich et al., 2003; Hunt et al., 1985; Xing-Sheng et al., 1983) and VHF radar estimates (and comparison with theoretical models) of refractive index structure function $C_n^2$ (VanZandt et al., 1978; Woodman and Guillen, 1974) have established the intermittent nature of turbulence within deep layers. More recently, quantitative aircraft measurements of turbulence kinetic energy dissipation rate $\epsilon$ and the temperature structure function $C_T^2$ have characterized the small-scale turbulence features within shallow turbulent layers in the troposphere (Balsley et al., 2018; Eaton et al., 1998; Fernando et al., 2015; Muschinski et al., 2001b; Scipión et al., 2016).

Various explanations for the prevalence of S&L structures have been proposed. Concurrent observations using VHF radars and in-situ measurements suggest that S&L are the result of multiscale gravity waves (GWs) interacting with the fine structure (FS) of the background atmosphere (Barat, 1982; Coulman et al., 1995; Luce et al., 1995; Röttger, 1980). Some analytical studies and numerical modeling results support this conjecture (Fairall et al., 1991; Fritts and Rastogi, 1985; Fritts et al., 2009a; Fua et al., 1982; Sidi et al., 1988; Smith et al., 1987; VanZandt et al., 1978). More recent Direct Numerical Simulations (DNS) achieving very high spatial and temporal resolution, primarily addressing multiscale GW and fine structure (GW-FS) interactions in "stable" environments (Fritts and Wang, 2013; Fritts et al., 2009b, 2013), suggest that Kelvin-Helmholtz instabilities (KHI), GW breaking and intrusions lead to the formation of S&L. On the other hand, Tjernström et al. (2009) suggested that airflow over low-relief terrain (i.e., small-scale mountain waves) are a plausible formation mechanism for S&L in the lower troposphere. Further work is needed combining more extensive observations with high-resolution modeling to understand the mechanism underlying S&L formation and evolution.

Initial modeling exploration of formation mechanisms of S&L structures arising from superposition of convectively stable GWs and dynamically stable mean shears, collectively referred to as multiscale dynamics (MSD), employed an idealized high resolution DNS (Balsley et al., 2018; Fritts et al., 2013; Fritts and Wang, 2013). The initial DNS of MSD by Fritts et al. (2013) featured a dynamically stable monochromatic GW of amplitude $a = (d\theta/dz)_{min}/(d\theta/dz) = 0.5$ and an intrinsic frequency $\omega = N/10$. A constant mean stability $N$, and $Re = 50,000$ were shown to enable instabilities and turbulence structures accompanying GW-FS dynamics that extend to very small-scales (Balsley et al., 2018; Fritts and Wang, 2013; Fritts et al., 2013). The DNS identified KHI evolving along the most highly stratified vortex sheet initiated by a propagating GW, that was

intensified by its upward wave displacement causing the local Richardson number to decrease below 0.25. This resulted in shallow turbulent layers laminated by thin stable layers resembling the S&L structures seen in VHF radar observations. Collectively, the S&L formation mechanism hypothesized by the initial MSD DNS and the advancing in-situ turbulence measurement capabilities of UAS provided the motivation for the IDEAL observation program.

Until recently, deeper understanding of the formation, morphology, and evolution of S&L and associated small-scale, weak, intermittent turbulence structures has been hampered by current turbulence observational methods that are limited by spatial and temporal resolution, and inadequate range and dexterity of measurement platforms (Chimonas, 1999; Muschinski et al., 2001a; Muschinski and Wode, 1998; Tjernström et al., 2009). Additionally, the single-point vertical profiles (instrumented towers, balloon borne soundings, and tethered lifting systems) provide little information about the lateral scales of S&L structures (Muschinski and Wode, 1998).

Although estimation of turbulence dissipation rate has been obtained from relatively high-resolution radiosonde measurements in the troposphere and lower stratosphere environments (Clayson and Kantha, 2008; Gong and Geller, 2010; Wilson et al., 2011; Kohma et al., 2019), recent advances in sensing abilities of UAS have enabled higher resolution, higher cadence, variable-path turbulence observations in the lower troposphere. Proof of concept turbulence measurements using UAVs such as the MMAV (van den Kroonenberg et al., 2008), MASC (Wildmann et al., 2014), BLUECAT (Witte et al., 2016), SUMO (Bäserud et al., 2016), Skywalker X6 (Calmer et al., 2018), ALADINA (Altstädter et al., 2015), and OVLI-TA Alaoui-Sosse et al. (2019) have been provided through integration of high-cadence fine-wire and multi-hole pressure probe sensors and deployment in various field campaigns limited to characterize turbulence in the boundary layer, e.g., CASES-99 (Balsley et al., 2003), MATERHORN (Fernando et al., 2015), and BALLAST (Bäserud et al., 2016; Båserud et al., 2014). Even so, the limited lateral-scale characterization and the dearth of high-resolution, quantitative measurements of turbulence parameters provide poor guidance for modeling studies employing high-resolution DNS investigating the S&L formation mechanisms.

The Instabilities, Dynamics, and Energetics accompanying Atmospheric Layering (IDEAL) project was conceived to address the current limitations in our understanding of the morphology and energetics of S&L turbulence through a synergistic combination of precisely targeted multipoint observations using small unmanned aircraft systems (sUAS) guiding DNS modeling to characterize the dynamics driving S&L structures and associated flow features. The first phase of the project featured an observational field campaign to systematically probe stable lower atmospheric conditions using multiple DataHawk2 sUAS (DH2) developed at the University of Colorado, guided by NCAR Integrated Sounding System (ISS) continuous radar profiling and hourly radiosonde profiling in and above the nocturnal boundary layer. Measurements were conducted employing multiple DH2, most commonly in sorties of three aircraft, for in-situ profiling and horizontal and/or slant path sampling. A total of 70 DH2 flights coordinated with 93 balloon-borne radiosondes were deployed supporting the IDEAL field campaign. Additionally, an array of Surface Atmospheric Measurement Systems (SAMS) collected surface winds, temperature, and relative humidity at 2 m (mini-SAMS) and 10 m (SAMS) to monitor surface and boundary layer activity. Observation locations of IDEAL field measurements are shown in Figure 1. Following the field campaign, the second phase focused on high-resolution DNS modeling efforts, guided by the in-situ observations, to permit more quantitative exploration of S&L morphology, energetics, and evolution.

90  This article focuses on the details of the observational phase of IDEAL: a campaign between 24 October and 15 November 2017 at the Dugway Proving Ground (DPG), Utah. Section 2 outlines the observation platforms used and the meteorological conditions during the campaign. Section 3 describes the UAS and radiosonde measurement strategy employed during IDEAL. Section 4 briefly outlines the data processing and the analysis techniques employed. Finally, section 5 provides concluding remarks and the scope for future work.

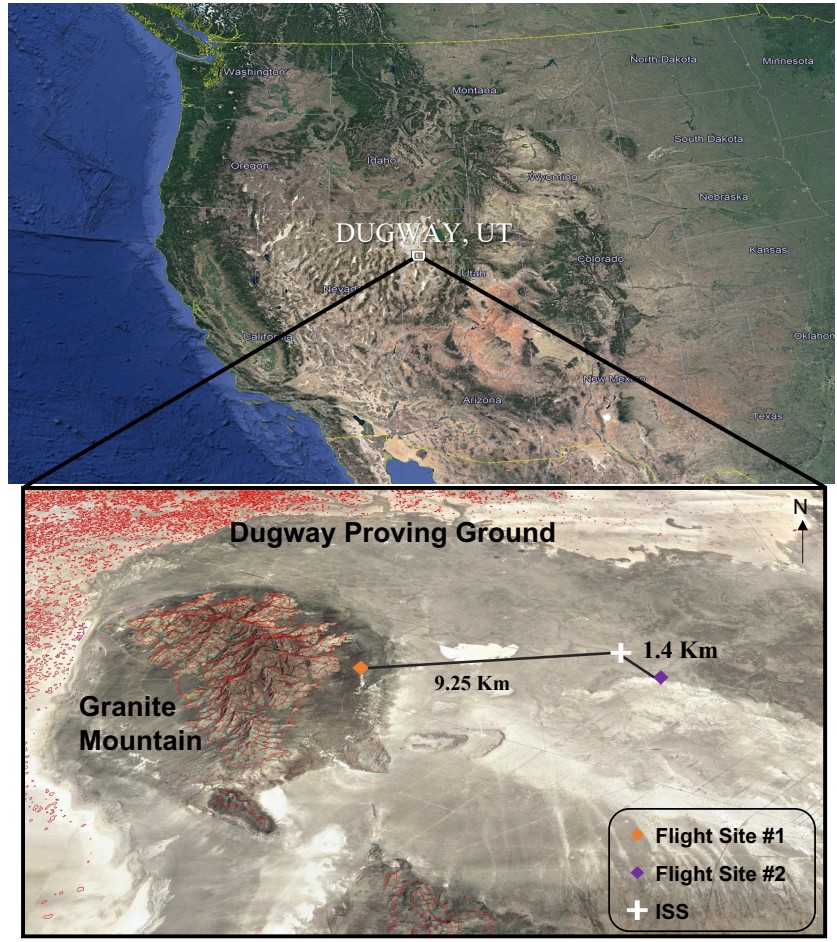

**Figure 1.** IDEAL field campaign location at DPG, Utah (top). The satellite image shows the predominantly flat DPG landscape (at 1320 m MSL) along with the prominent Granite Mountain feature (800 m AGL peak height). Locations of UAS launch sites (orange and purple diamonds) and the ISS deployment site (white cross) are shown. Satellite imagery was obtained from the publicly available ©Google Earth 3D mapping tool.

## 2 Measurement Platforms and Observed Meteorological Conditions

### 2.1 DataHawk sUAS

The DataHawk2 sUAS (DH2) used for IDEAL measurements (see Figure 2, and Tables 1 & 2) is a product of many years of development at the University of Colorado. It is specifically designed for making high-resolution in-situ observations in the lower troposphere, and for operations in challenging surface conditions. The precursor DH1 was used in campaigns in Peru (Balsley et al., 2013; Lawrence and Balsley, 2013; Scipión et al., 2016) and Utah (Balsley et al., 2018; Fernando et al., 2015). The DH2 was used in campaigns in Japan (Kantha et al., 2017, 2019; Luce et al., 2018a, b, 2019, 2017), Colorado (de Boer et al., 2019), and Alaska (de Boer et al., 2018). To date, over 650 science flights have been performed with the DH2, totaling 430 flight hours. Relevant attributes of DH2 sUAS as they relate to the IDEAL field program are noted below:

- **Low cost.** At approximately $1,000 each, many vehicles can be deployed for a campaign, enabling multiple simultaneous measurements (as employed extensively for IDEAL) or sequences of overlapping flights to provide continuous measurements over many hours. This also enables observations in marginal conditions (e.g., high winds) that would ground more expensive vehicles due to the risk of loss. Ten DH2 vehicles were constructed for the 23-day IDEAL campaign.

- **Ruggedness.** The airframe is resilient foam, strengthened by a system of interior spars and flexures that absorb impacts, enabling the vehicle to "bounce" rather than break when landing on unprepared surfaces. It has a no-tail design, since these extended members are easily broken, and resilient wing trailing edges and vertical fins that are very difficult to break. It also has a rear propeller with folding blades to prevent damage to the propulsion system during landing. In the IDEAL campaign, five DH2 aircraft were used extensively, of which two were retired due to accumulated wear. No aircraft were lost.

- **Ease of operation.** A custom autopilot provides automatic launch, landing, and vector field flight control (Lawrence et al., 2008), enabling a variety of measurement strategies to be set up with ease and flown under minimal operator supervision. Flight patterns can also be changed during flight to target specific volumes of interest, e.g., based on real-time measurements—an ability that was extensively used during IDEAL to identify and more thoroughly sample turbulent layers. A bungee cord is used for launch, guided by a simple two-rail launch ramp (see Figure 2).

- **Gust-insensitive design.** The unique aerodynamic design eliminates the roll moment due to sideslip, making the vehicle point into gusts rather than roll away from them, enabling well-behaved flight in high-wind and strong turbulence conditions. Normally, flights are not performed when surface winds exceed $10 \ \mathrm{ms^{-1}}$, or predicted winds aloft exceed $15 \ \mathrm{ms^{-1}}$. The vector field guidance uses a wind-aware algorithm that tracks derived compass heading to stabilize flight even when wind speed exceeds airspeed, eliminating the "reverse course" behavior that occurs in this case when GPS heading is tracked. During IDEAL, synoptic winds above 3000 m typically exceeded $20 \ \mathrm{ms^{-1}}$ which limited the flight ceiling to this altitude to prevent unrecoverable downwind drift.

- **Flexible sensor interfacing.** The custom DataHawk autopilot provides multiple serial interfaces (7 UART, 3 I2C, 4 SPI), enabling a variety of sensors to be supported, and their data stored on-board (on a micro SD card), and telemetered to the ground station for real-time display. Tables 1 and 2 provide details of the sensors employed for IDEAL. Sensors can be installed at various locations in the body or the wings of the airframe without altering the flight dynamics, provided center-of-mass constraints are preserved.

- **Efficiency.** Flight durations exceed 80 min nominally, making altitudes of 5 km above a ground launch accessible with a typical 2 $ms^{-1}$ ascent/descent rate, and a lateral range (out and back) of 30 km at a nominal airspeed of 15 $ms^{-1}$. For IDEAL, this enabled profiles to 3000 m AGL plus extra time exploring interesting layers for the vertically sampling aircraft, and many km-long racetrack patterns to be traversed by the laterally sampling aircraft.

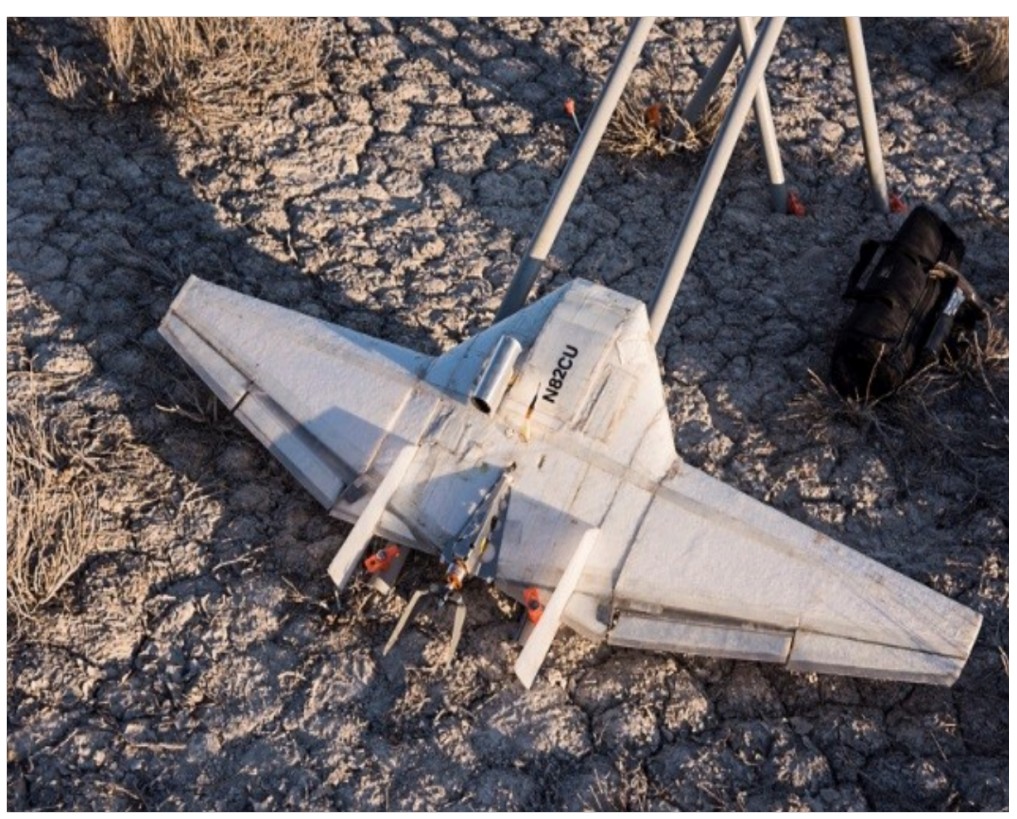

**Figure 2.** DH2 sUAS ready for launch at DPG during the IDEAL campaign.

For IDEAL, the DH2 was configured to make the following in-situ observations. Characteristics of these observations are summarized in Table 2.

1. **Measurement location and time.** A UBlox M8N single-frequency GPS receiver provides horizontal position data and time at a 5 Hz cadence. Horizontal position typically wanders within a 10 m error band at a stationary location. Al-

titude measurement is refined in post-flight analysis to obtain high-vertical resolution by calibrating the higher rate of response (100 Hz) but slowly drifting barometric pressure altitude (using the TE MS5611 sensor) against the low-rate ($\sim 5$ Hz) GPS altitude, that does not have long-term drift but can shift occasionally as different satellites come into view. The resulting resolution in altitude is dominated by the random noise level in the pressure altitude of approximately 20 cm RMS. Similarly, sensor measurement times are recorded at high-resolution by calibrating 10 microsecond micro-processor timer ticks to 5 Hz GPS time of week (TOW) data in post-flight time-alignment procedures. The resulting time-alignment lies within an uncertainty of 0.2 s.

2. **Mean atmospheric state.** In-situ temperature and relative humidity (T/RH) are provided by a Sensirion SHT-31, located in the flow stream inside a cylindrical shroud, mounted above the vehicle as shown in Figure 2. Manufacturer's specifications for T/RH are quoted in Table 2. Barometric pressure is provided by a TE MS5611 absolute pressure sensor, located in an unsealed compartment within the foam aircraft body, with manufacturer's specifications given in Table 2.

3. **High-resolution temperature.** A custom (coldwire) thermometer uses a five-micrometer diameter platinum wire to measure mean flow temperature as well as fine-scale temperature variations in the flow. With a time constant of 0.5 millisecond and a sampling rate of 800 Hz, temperature variations at wavenumbers of $\sim 20$ m$^{-1}$ can be detected at the nominal 15 ms$^{-1}$ airspeed. The high-resolution temperature measurement is calibrated against the collocated (but slow) SHT temperature (in $^\circ$C) by employing linear least squares regression in post flight analysis. Uncertainty in this calibration depends on the temperature range observed: with the vertical profiles to 3 km AGL in IDEAL providing a temperature range of 20 $^\circ$C, the uncertainty in the SHT temperature results in mean coldwire temperature uncertainty of 0.2 $^\circ$C and an uncertainty in coldwire temperature fluctuation scale factor of 5%. The spectral noise floor of coldwire temperature produces RMS noise in flight of 0.002 $^\circ$C. Calibrated coldwire temperature is used with high-resolution altitude to obtain high-vertical resolution potential temperature $\theta$. Over several km altitude change, uncertainty in $\theta$ depends primarily on uncertainty in temperature, leading to the $\theta$ characteristics shown in Table 2, assuming a surface temperature of $T_o$. Spectral analysis is also used to fit inertial sub-range power spectral density models to provide estimates of the turbulent temperature structure parameter $C_T^2$ (Frehlich et al., 2003; Luce et al., 2019). The accuracy of the resulting turbulence parameters is a function of the quality-of-fit of the spectral data to the Kolmogorov cascade model; misfits can be due to many causes, and various measures have been proposed (see e.g., Luce et al. (2019)). Also, the turbulence intensity parameterized by $C_T^2$ covers six orders of magnitude, so accuracy is best characterized logarithmically as a fraction of a decade. The accuracy values quoted in Table 2 correspond to cases with the highest-quality fits observed in IDEAL data.

4. **High-resolution airspeed.** A custom pitot-static tube and a TE MS4515 differential pressure sensor provide 800 Hz airspeed data that is calibrated to ms$^{-1}$. Coarse calibration is provided by relating measured dynamic pressure to air-speed, accounting for density and a nominal calibration factor found from wind tunnel testing with known airspeeds. This is used for flight control. Fine calibration of Pitot velocity is conducted in post-flight analysis using portions of the flight that have circular helical trajectories. These enable GPS horizontal velocity excursions to determine adjustments

to the calibration factor accurate to 0.2 m/s. Offsets in this calibration are removed by averaging Pitot data during pre-flight procedures where the Pitot tube is covered. Motor/propeller vibrations and signal quantization produce an in-flight resolution (noise floor) of $0.07$ $\mathrm{ms}^{-1}$ RMS that increases at higher throttle settings. Wavenumber resolutions similar to temperature fluctuations are obtained in velocity variations also, and similar spectral estimation methods are used to derive turbulent kinetic energy dissipation rate $\epsilon$ from the Pitot flucations (Frehlich et al., 2003; Luce et al., 2019). Filtered airspeed data are also used to estimate winds (described below). In addition, a custom (hotwire) anemometer uses a second five-micrometer diameter platinum wire to detect fine-scale velocity variations (calibrated against Pitot airspeed), and these are also used to estimate $\epsilon$, but at a higher confidence level due to the absence of motor vibration artifacts that typically appear in the Pitot velocity spectra at high frequencies. As for $C_T^2$ above, accuracy levels quoted for $\epsilon$ in Table 2 derive from the highest-quality fits of hotwire spectral data with the Kolmogorov inertial sub-range model found in IDEAL data.

5. **Horizontal Wind.** Vehicle GPS velocity is combined with Pitot airspeed and vehicle attitude to produce estimates of the horizontal wind at 1 Hz cadence in post flight analysis, using an extension of the geometrical method from (Lawrence and Balsley, 2013) that provides higher-resolution wind estimates. A detailed description of this approach is in preparation for publication. Since wind estimation from a moving vehicle involves multiple sensors, often with differing time constants and error characteristics, and comparison with reference sensors (e.g. on fixed towers) is problematic, accuracy of wind sensing is elusive. The value given in Table 2 is based on the magnitude of remaining anomalies in wind retrievals over circular helical profiles where periodic artifacts are relatively easy to spot. Since there are various ways to extract wind estimates from the IDEAL data, the wind estimates provided in the IDEAL data set are intended only for "quick look" purposes in surveying the conditions for each flight. Users may want to develop their own wind estimation products to suit the analysis at hand.

6. **Atmospheric stability.** The Brunt-Vaisala (buoyancy) frequency $N$ is evaluated using the vertical gradient of high-resolution potential temperature $\theta$ using high-resolution altitude. This becomes highly uncertain as the vertical displacement in a finite difference for the gradient becomes small, so uncertainty estimates depend on the vertical step sizes used, as well as any smoothing for $\theta$ and altitude signals used in this estimate. Estimates of $N^2$ are provided in the IDEAL data set for purposes of highlighting the structure of background stability; users of the data set may want to develop $N^2$ estimates specific to their needs.

7. **Forcing conditions.** Horizontal wind shear forcing is assessed relative to the background layer stability via the gradient Richardson number Ri, derived from the horizontal mean wind gradient with altitude, and the local buoyancy frequency. Again, uncertainty in this estimate depends on the details of the gradient estimation process, hence is user-dependent. Quick look plots of Ri are provided in the archived IDEAL data, but are meant only as guides to features in the data. Users may wish to estimate Ri and associated uncertainty as needed for their analyses.

| DH2 Characteristics | | DH2 Capabilities | |
|---|---|---|---|
| Wingspan | 1.3 m | Airspeed | 10-20 ms$^{-1}$ |
| Mass | 1.3 kg | Duration | 80 minutes |
| Vehicle Cost | $1000 | Range (one way) | 60 km |
| Sensor Cost | $400 | Altitude (balloon Drop) | 6 km AGL |
| Design | Flying wing, rear propeller | Altitude (ground launch) | 5 km AGL |
| Telemetry | XBee radios at 2.4 GHz or 900 MHz | Turning radius | $> 50$ m |
| Propulsion | Electric, folding propeller | Climb rate | $< 3$ ms$^{-1}$ |
| Autopilot | Custom M4 | Downlink throughput | $> 1500$ bytes per second |
| Control | Auto, operator supervised | Downlink update rate | 10 Hz |
| Power | 11 V LiPo, 7600 mAhr | Sensor sampling | up to 800 Hz |
| Construction | Polypropylene foam | Data storage (on board) | Micro SD card |

**Table 1.** Characteristics of the DH2 sUAS as outfitted for the IDEAL campaign.

| Type | Resolution | Accuracy Range | Time Constant Cadence | Notes |
|---|---|---|---|---|
| Hor. Location (GPS) | 10 cm | 10 m; worldwide | 0.2 s; 5 Hz | Real time |
| Altitude | 20 cm | 1 m; -1 km to 20 km MSL | 1 ms; 100 Hz | Post flight calibration |
| Time (GPS) | 1 ms | 0.2 s; 1 week | 0.2 s; 5 Hz | Real time |
| SHT temperature | 0.01 °C | 0.2 °C; -60 °C to +40 °C | 2 s; 10 Hz | Real time |
| Relative humidity | 0.01 % | 4 %; 0 % to 100 % | 8 s; 10 Hz | Real time |
| Barometric Pressure | 0.012 mbar | 5 mbar; 450 to 1200 mbar | 8.2 ms; 100 Hz | Real time |
| Airspeed | 0.07 ms$^{-1}$ | 0.2 ms$^{-1}$; 10 ms$^{-1}$to 20 ms$^{-1}$ | 5 ms; 800 Hz | Post flight calibration |
| Coldwire temperature | 0.002 °C | 0.2 °C; -60 °C to +40 °C | 0.5 ms; 800 Hz | Post flight calibration |
| Potential temp. $\theta$ | 0.002 K | 0.2 K; $T_o$ to $T_o + 30$ K | 0.5 ms; 800 Hz | Post flight calibration |
| Hotwire velocity | 0.01 ms$^{-1}$ | 0.2 ms$^{-1}$, 10 ms$^{-1}$ to 20 ms$^{-1}$ | 0.5 ms; 800 Hz | Post flight calibration |
| Temp. Struc. Param. $C_T^2$ | $10^{-6}$ m$^{-2/3}$K$^2$ | 0.2 decade; $10^{-6}$ to 1.0 m$^{-2/3}$K$^2$ | N/A; 1 Hz | Post flight calibration |
| TKE Diss. Rate $\epsilon$ | $10^{-7}$ m$^2$s$^{-3}$ | 0.2 decade; $10^{-7}$ to 0.1 m$^2$s$^{-3}$ | N/A; 1 Hz | Post flight calibration |
| 2D vector wind | 0.05 ms$^{-1}$ | 0.5 ms$^{-1}$; 0 ms$^{-1}$ to 30 ms$^{-1}$ | 0.1 s; 1 Hz | Post flight calibration |

**Table 2.** Sensing Capabilities of the DH2 sUAS in the IDEAL campaign.

## 2.2 Integrated Sounding System (ISS)

An Integrated Sounding System (Parsons et al., 1994) was deployed to monitor the large-scale wind and thermodynamic environment, in proximity to the sUAS measurements. The ISS consisted of a Vaisala MW41 radiosonde sounding system, a LAP3000 915 MHz radar wind profiler, and Lufft WS700/WS800 surface meteorological sensors on a mast at 2 and 10 m. Ninety-three balloon-borne RS41-SGP radiosondes were launched in total, with daily launches between 0300 AM and 0700 AM LT at 30 to 60 min intervals, providing five to nine soundings each measurement day.

The ISS automatically ingests surface observations from the set of reference sensors (T/RH, and wind using Lufft WS700 and pressure using Vaisala PTB210) at 1.8 and 3 m. To achieve frequent radiosonde soundings ($< 60$ min apart), communications were terminated (at 12 km AGL) well before balloon burst to enable launch preparations for subsequent soundings. The balloon Helium volume was adjusted to achieve a median ascent rate of $\sim 3.5$ ms$^{-1}$.

The LAP3000 915 MHz radar wind profiler was operated in a low-height range mode to provide data at 60 m intervals between 200 m to 4.5 km AGL. Due to the dry conditions, winds were measured only up to 2 km on most days. The radar employed five beam directions and raw Doppler spectra were recorded every 30 s. Eastward and northward wind components were calculated from spectral moments averaged over 30 min.

Time-altitude data from radiosondes and the wind profiler were relayed hourly to the sUAS flight deployment team to aid in-flight planning. Examples of ISS observations on $6^{th}$ November 2017 are shown in Figure 3. Wind profiler data were used to monitor relevant events like precipitation (descending features in signal-to-noise ratio (SNR)), low-level jets (midnight at $\sim 2$ km), convective instabilities, and KHI.

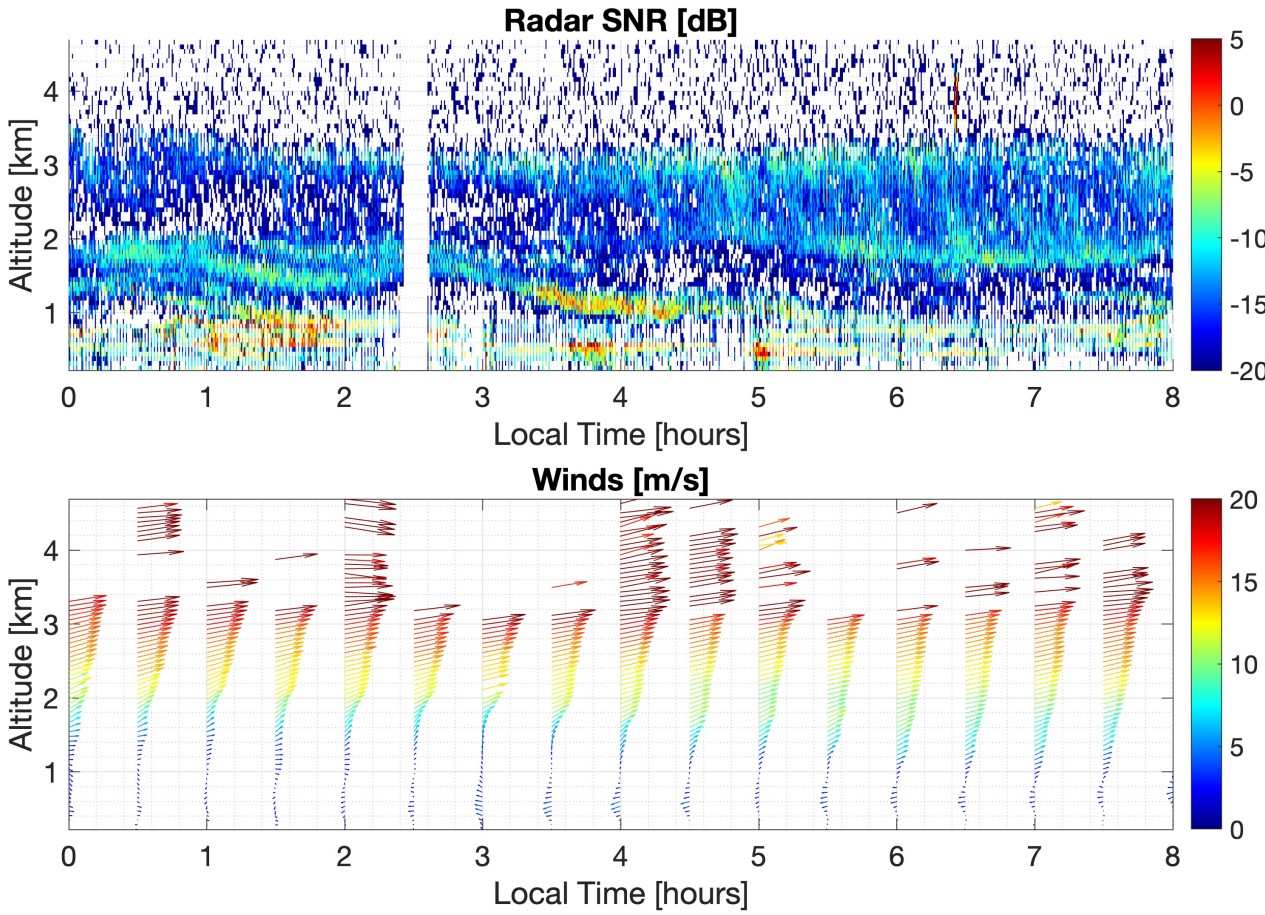

**Figure 3.** Example measurements by the LAP3000 915 MHz radar wind profiler showing SNR (top) and wind barbs (bottom) for the 6th of November from 0000-0800 LT. The radar is briefly turned-off periodically for maintenance - seen as a pause between 0200-0300 LT.

### 2.3 Campaign Meteorological Conditions

Weather forecasts for sUAS flight planning were provided by the DPG Meteorology Division. Weather briefings consisted of 2-day forecasts from the Four-Dimensional Weather System (4DWX), and the most recent observations of surface and upper-level conditions obtained from the DPG MET instrumentation. 4DWX uses the advanced research version of the Weather Research and Forecasting (WRF) model to predict the weather conditions at the US Army Test and Evaluation Command (ATEC) ranges (Knievel et al., 2017; Liu et al., 2008). The system is a product of collaboration between ATEC and NCAR. The local surface conditions were obtained using a network of towers that includes 31 SAMS and 50 mini-SAMS. Each SAMS reports 5 min averaged wind speed and direction at 2 m and 10 m, temperature, and relative humidity (T/RH) at 2 m, and precipitation. The mini-SAMS towers provide additional 10 m T/RH measurements with average values reported every minute. Doppler radar

wind profilers provided real-time wind profiles from 120 m up to 5 km. The forecasts included expected synoptic-scale wind patterns, expected times of frontal passage, development of surface inversions, and cloud cover.

During the campaign, DH2s were flown between 2 and 8 AM LT to sample the evolution of nocturnal atmospheric conditions. Weather briefings were provided to the team each day at 0:30 AM, so that launch sites and deployment strategies could be specified based on the most recent information. Observed T/RH and winds from all the soundings throughout the campaign are shown in Figure 4. Conditions were mostly dry with occasional evening precipitation. Surface winds during the first week of the campaign (24 October to 1 November 2017) were consistently strong from the South (see bottom right panel in Figure 4). Thereafter, surface winds were consistently from the North; the Northerly surface winds predicted in the valley for the last two weeks of the campaign agreed closely with the DPG MET 449 MHz wind profiler measurements. Predicted surface temperatures were between 0 and -5 °C for most nights.

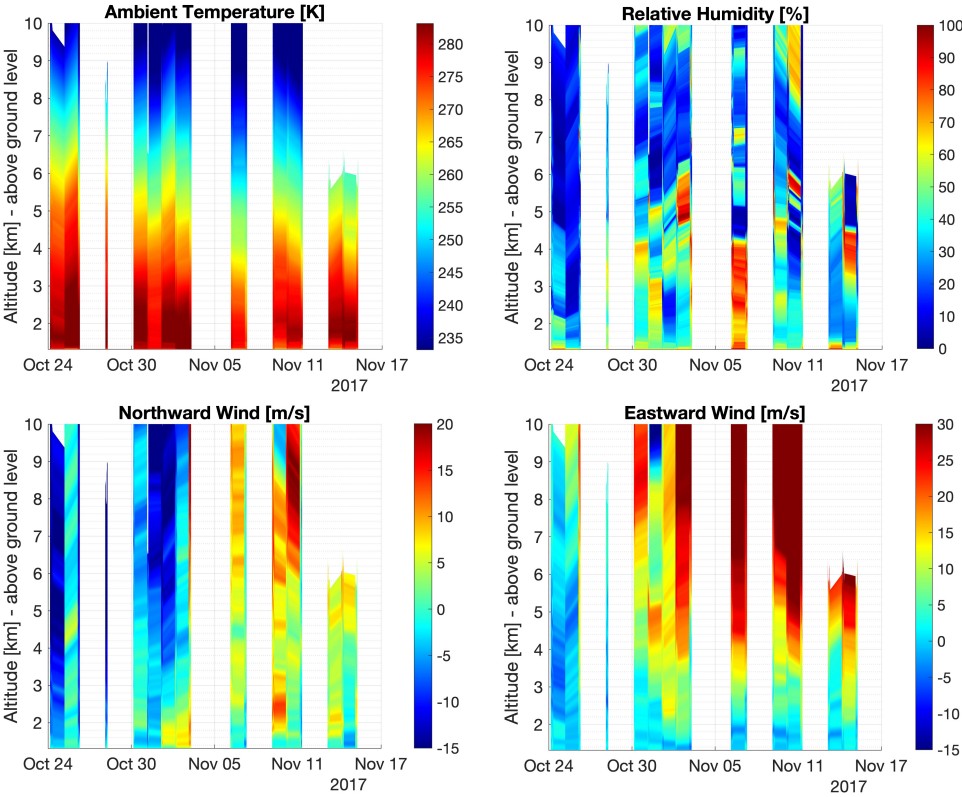

**Figure 4.** Vertical profiles of temperature (top left), humidity (top right), Eastward and northward winds (bottom: left and right) measured by 93 radiosondes deployed during IDEAL. Note: communications with some radiosondes were terminated early (e.g. $14^{th}$ November) to facilitate a faster launch cadence.

The stable nocturnal boundary layer was shallow on most nights ($\sim 75$ m), but occasionally increased to 200 m. Directional shear was frequently observed between the surface and altitude of 2 km. A weak westerly nocturnal jet was often observed at

2 km. Strong speed shears observed above 2 km resulted in infrequent KHI. The background atmosphere was statically stable

($N^2 > 0$) on most nights, with occurrences of stability and humidity sheets at various altitudes separated by weak, intermittent, and sporadic turbulence events.

## 3   Observational Strategy

The IDEAL measurement program was designed to take advantage of the high-spatial resolution, range, and dexterity of DH2 aircraft to provide high-resolution in-situ observations of S&L structures and their evolution under stable boundary layer and

lower troposphere conditions. The low cost and ease of operation of DH2 sUAS discussed in Section 2.1 enabled simultaneous multi-path measurements intended to quantify local S&L flow evolution, scales, and the dynamics underlying their small-scale structures. Because in-situ measurements are necessarily sparse, DH2 trajectories were designed to provide successive, multiple-DH2 sampling of local flows along horizontal, inclined, and spiraling vertical flight paths sampling common volumes over tens of minutes.

Flight planning relied predominantly on 4DWX weather forecasts and local ISS and SAMS measurements described in Section 2.2. Evolution of predicted winds and the thermodynamic state of the synoptic-scale flow were used to identify the likely most favorable site for DH2 measurements each day. The two flight operation sites were established as shown in Figure 1: one on the eastern flank of Granite mountain, marked 'Flight Site 1' (FS1), and one in the central portion of the valley, marked 'Flight Site 2' (FS2). FS1 and FS2 were chosen to be upwind of the ISS deployment site for two different wind conditions.

Flight operations were conducted from FS1 on days when the predicted surface winds were from the southwest or west, and the lower level forcing was at least as strong as 8 ms$^{-1}$ to 10 ms$^{-1}$. This maximized the likelihood of observing mountain wave influences on S&L structures due to Granite mountain at FS1, and accompanying fine-structure interactions leading to S&L dynamics. Flight operations were conducted from FS2 on days when the predicted winds were from the south or north, or were relatively weak.

Radiosonde winds and temperatures characterized the S&L structures at coarse vertical resolution which guided the choice of measurement location. An example of the real-time radiosonde data relayed to the measurement team on 6 November 2017 is shown in Figure 5. These periodic soundings monitored the spatial variability, intermittency, and temporal evolution of the layered structures at low vertical resolution and contributed to go/no-go decisions for DH2 flight sorties based on the prospects for encountering interesting dynamics while avoiding high-wind conditions.

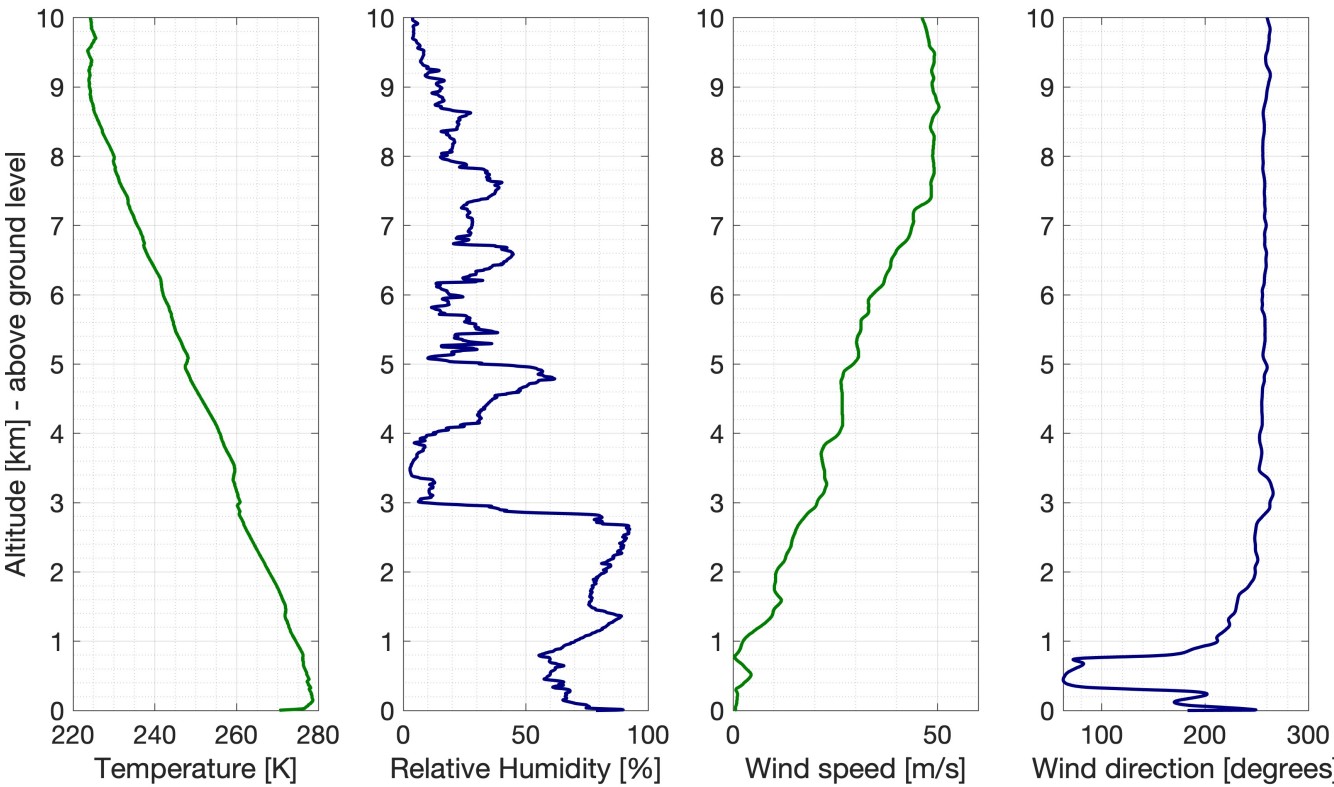

**Figure 5.** Altitude profiles of horizontal wind speed, direction, temperature, and relative humidity from a sample radiosonde launch at 0000 LT on 6 November 2017.

Over a span of 23 days, a total of 14 flight sorties from FS1 and 17 sorties from FS2 were performed in co-ordination with 93 periodically launched radiosondes. Rapidly changing atmospheric features around Granite Mountain influenced the choice of number of aircraft deployed in each sortie. Typical sorties during IDEAL contained between one and three DH2 aircraft each. Horizontal wind speeds above 3000 m frequently exceeded $15\ \mathrm{ms^{-1}}$ which limited the flight ceiling to this altitude. Figures 6 and 7 show the 3D contour of Granite Mountain overlaid with the DH2 flight trajectories for the two sorties carried out on

6 November 2017. The sorties shown in Figures 6, 7, and 8 sought to investigate the temporal evolution of multiple layers at different altitudes with measurements that were spaced evenly in time. A typical DH2 flight sortie during IDEAL consisted of one vertically sounding aircraft (such as A1 in figures 6, 7, and 8) flying a helical pattern of $100\ \mathrm{m}$ radius with nominal airspeed ranging between $14-18\ \mathrm{ms^{-1}}$ and ascending/descending between $1-4\ \mathrm{ms^{-1}}$. Each sortie also consisted of laterally sounding aircraft (such as A2 and A3 in figures 6, 7, and 8). The trajectories of lateral sounding aircraft in each sortie varied depending

on the conditions relayed by recent radiosonde data and on the conditions observed by A1. Generally, the lateral aircraft were directed to concentrate on a particular turbulent layer evolution. They followed horizontal racetrack patterns spanning about $1.5$ km, oriented with the racetrack long axis along the horizontal wind direction while slowly ascending and descending through a narrow altitude range to observe spatial and temporal variability in the layer.

Table 3 lists the date, flight launch site, launch time, flight ceiling and measurement strategy for each DH2 flight for five notable sorties from the IDEAL campaign. The atmospheric conditions during these sorties are given in the caption of Table 3. A complete list of all the DH2 flight sorties are provided in Table 4 in the appendix section. The overview plots of T/RH, static pressure, wind speed and direction, 3D GPS position, and aircraft velocity data along with the flight notes for each sortie are available for download on the IDEAL project preliminary analysis web page hosted by the University of Colorado (see Section 4).

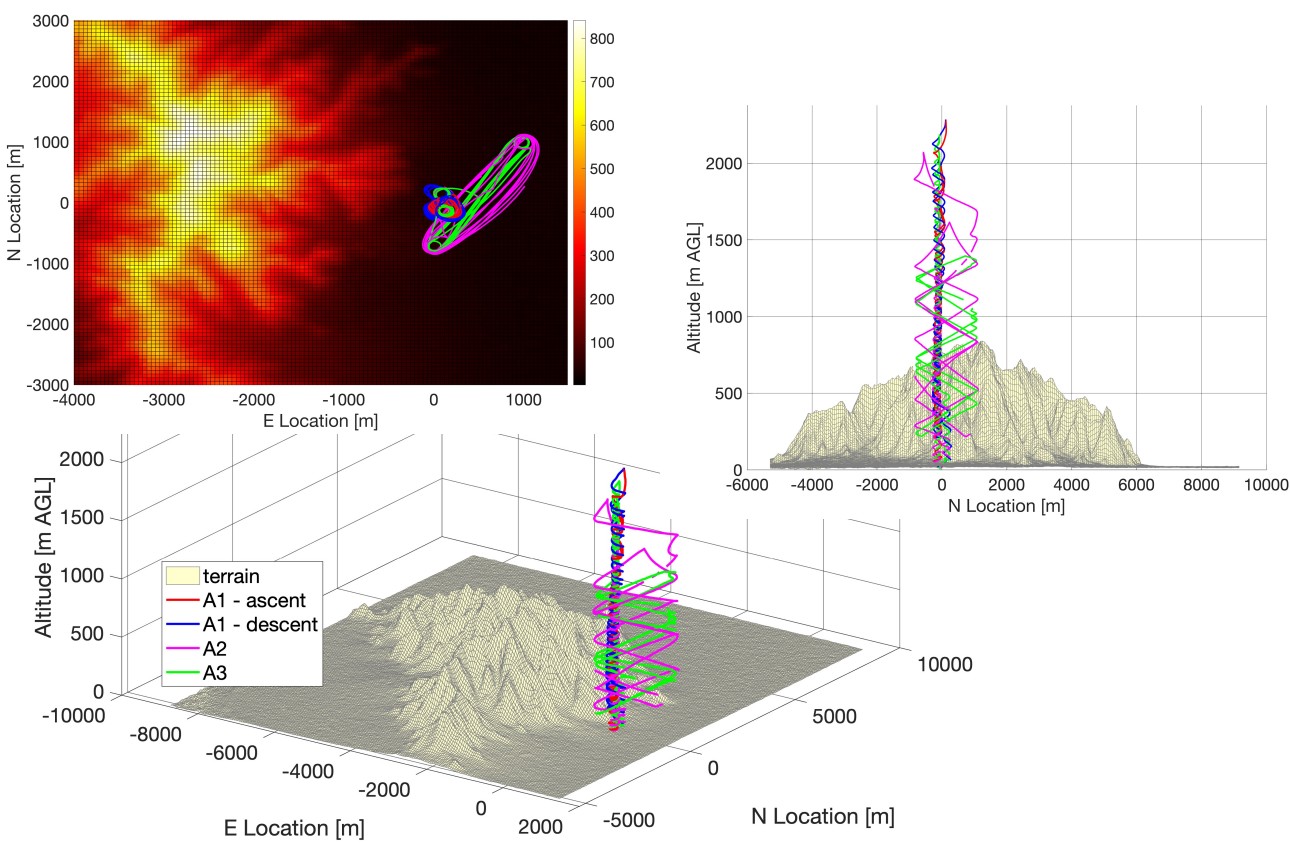

**Figure 6.** (Left and top right panels) Plots depicting the trajectories of the three aircraft A1, A2, and A3 deployed at FS1 for sortie 1 on 6 November 2017. A dominating terrain feature is Granite mountain (standing 850 m above the surrounding plane). (Bottom Right panel) Horizontal map showing Granite mountain with mean wind and DH2 trajectories.

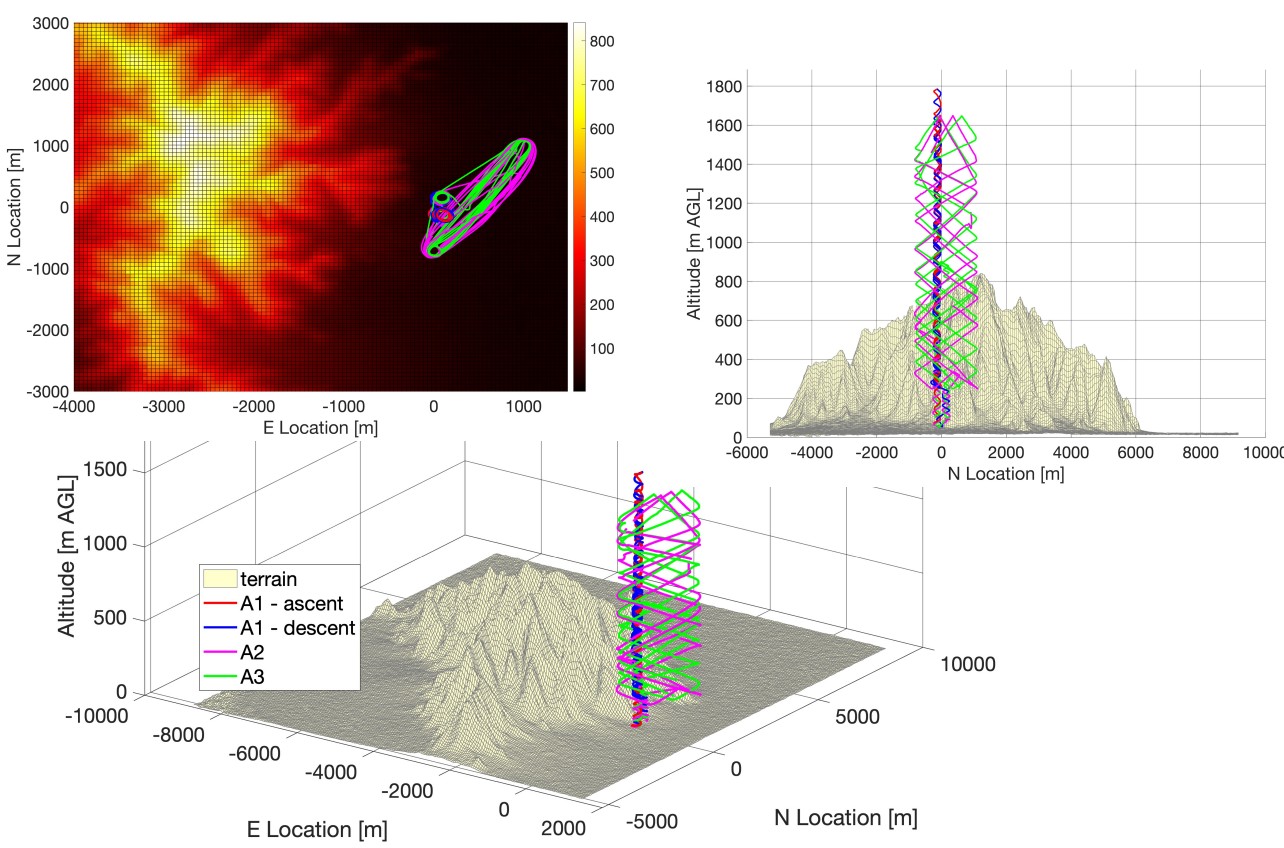

**Figure 7.** Trajectories of the vertically 'sounding' aircraft A1, and 'laterally' sounding aircraft A2 and A3 from Sortie 2 on 6 November 2017.

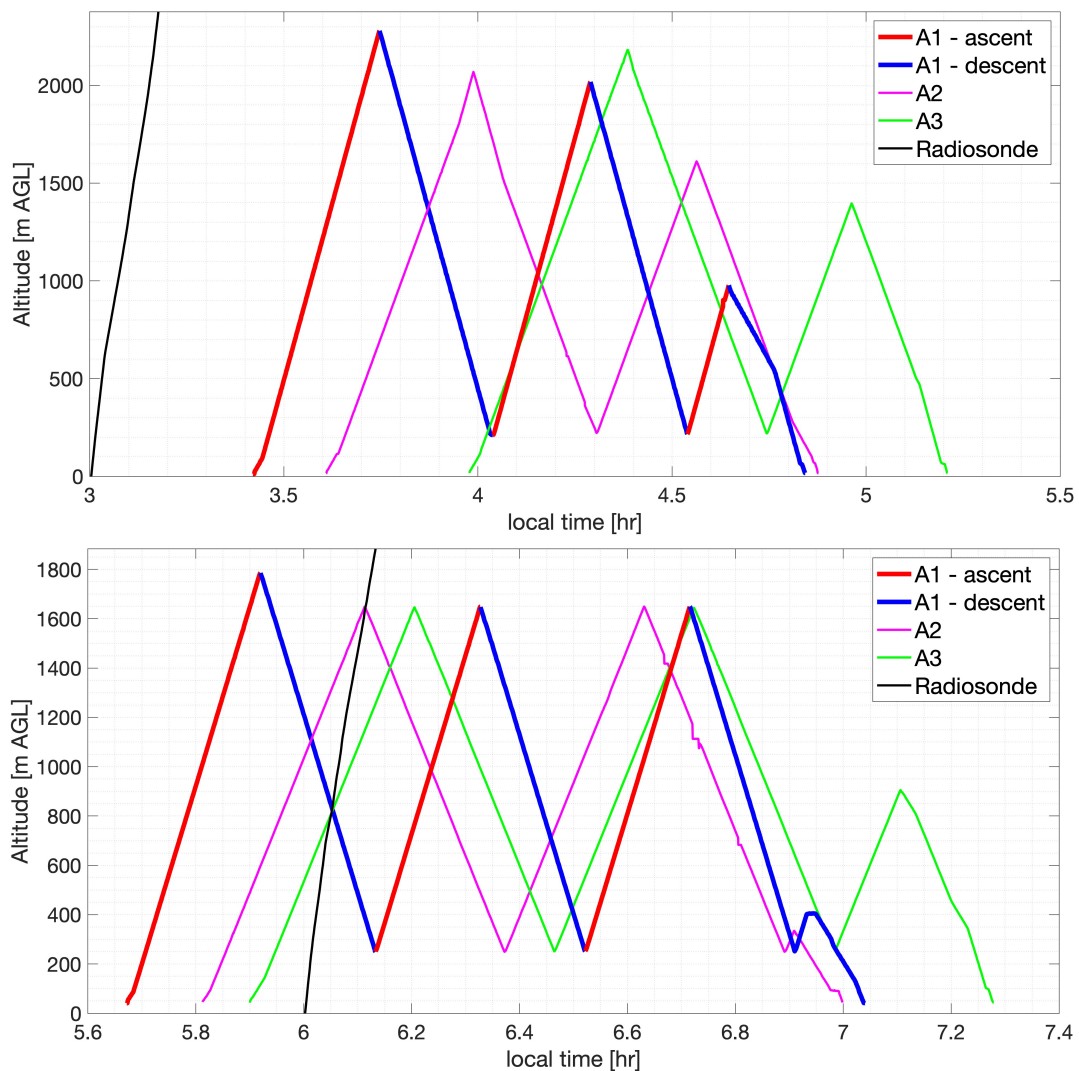

**Figure 8.** Timeseries (LT) showing altitude profiles for vertically sounding aircraft A1 (red - ascent; blue - descent), and laterally sounding aircraft A2 and A3 (magenta and green) co-ordinated with a radiosonde trajectory (black) for sorties 1 (top) and 2 (bottom) on 6 November 2017.

| Sortie number | Date | Flight launch time (HH:MM) | Flight launch site | Flight number | Meas. strategy and target alt. |
|---|---|---|---|---|---|
| S08 | 01-Nov-17 | 03:16 | FS1 | DH16 DH17 | DH17 - spiral; ceiling - 3100 m DH16 - slant racetrack between 200 - 2700 m |
| S11 | 02-Nov-17 | 05:12 | FS2 | DH22 DH23 | DH22, DH23 - slant racetrack between 50 - 2600 m |
| S17 | 06-Nov-17 | 05:17 | FS1 | DH34 DH35 DH36 | DH36 - spiral; ceiling - 1800 m DH34, DH35 - slant racetrack between 200 - 1600 m |
| S23 | 10-Nov-17 | 02:53 | FS1 | DH46 DH47 DH48 | DH48 - spiral; ceiling - 2600 m DH46, DH47 - slant racetrack between 200 - 2500 m |
| S29 | 14-Nov-17 | 02:53 | FS2 | DH63 DH64 DH65 | DH65 - spiral; ceiling - 2200 m DH63, DH64 - slant racetrack between 100 - 1400 m |

**Table 3.** List of five notable DH2 UAS sorties deployed during IDEAL observation campaign. The launch date, site, time (HH:MM, LT), flight ceiling and observation strategy for aircraft in each sortie are presented. Sorties S08 and S17 observed persisting S&L structures at FS2. S23 sortie conducted measurements in a quickly dissipating turbulent layer at FS2. Sorties S11 and S29 sampled the nocturnal BL and the residual layer turbulence at FS1.

## 4  Data Processing and Preliminary Results

The sensor data sampled from DH2 observation flights was telemetered to the ground station for real-time display and also written periodically on-board a micro SD card as binary data in $4$ KB packets. An extensive suite of programs has been developed using MATLAB to calibrate the raw DH2 sensor measurements and compute meaningful scientific data products during post-flight data analysis. The DH2 instrument configuration, outlined in Section 2.1, consists of a custom Pitot static tube fitted to a pressure sensor, a hotwire anemometer, and a coldwire thermometer that provide airspeed (in $\mathrm{ms}^{-1}$) and temperature (in K) at a high-cadence of $800$ Hz that are used to estimate $\epsilon$ and $C_T^2$. The Pitot derived airspeed data along with the GPS velocity and aircraft attitude information provided by the DH2 autopilot standard sensor suite are used to estimate the mean horizontal wind vector. A novel wind estimation algorithm, improving the estimation procedure described by Lawrence and Balsley (2013), is described in detail in Doddi (2021) (doctoral thesis). This provides estimates of the horizontal wind vector at a cadence of $1$ Hz and at an accuracy up to $0.5 \mathrm{~ms}^{-1}$.

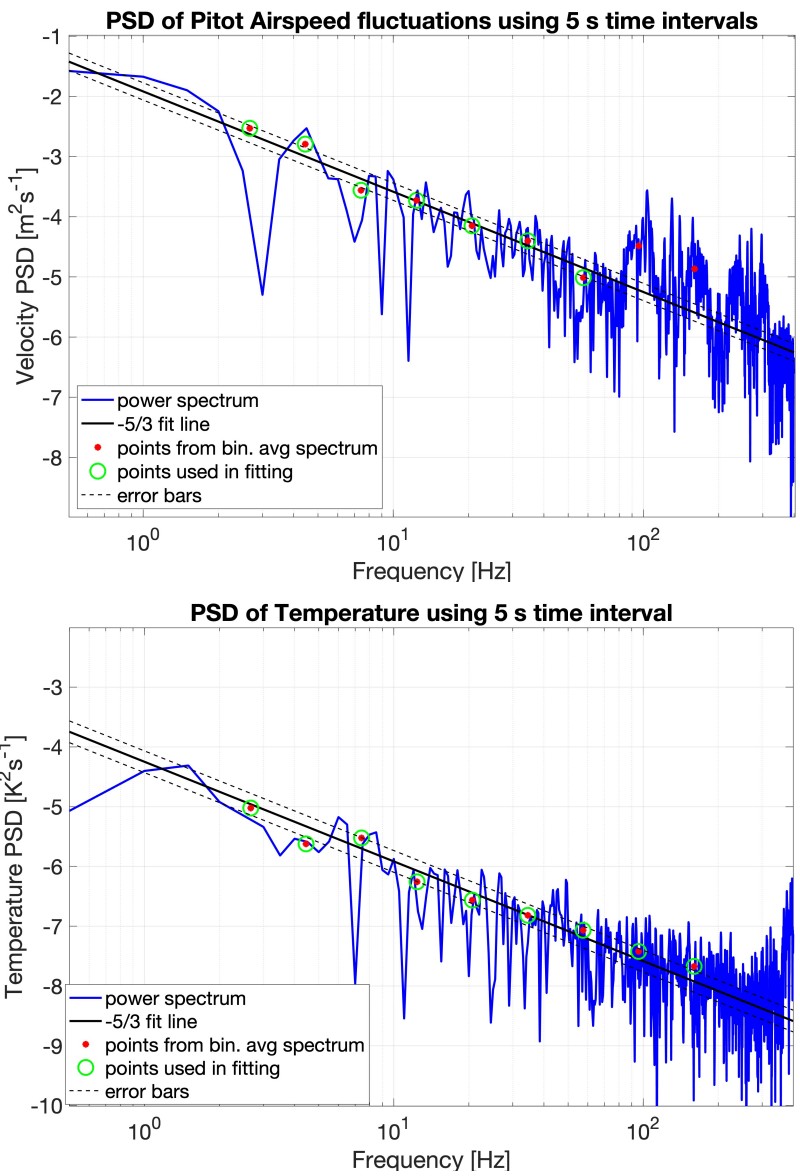

**Figure 9.** PSD of Pitot derived airspeed (top tile) and coldwire derived temperature (bottom tile) plotted against sampling frequency. The $f^{-5/3}$ fit line (solid black) along with the fit variance (dashed black) are shown.

Following the estimation procedures described by Frehlich et al. (2003) and Luce et al. (2019) the turbulence parameters of $\epsilon$ and $C_T^2$ are determined by applying spectral analysis to the high-cadence airspeed and temperature measurements. The power spectral density (PSD) is computed using non-overlapping $5$ s time intervals. These raw PSD are then fit to the model spectra proposed by Tatarskii (1971) using a novel spectral fitting algorithm, then converted to turbulence parameters $\epsilon$ (for velocity) and $C_T^2$ (for temperature). Details of the spectral analysis and fitting procedures are also described in detail in Doddi (2021).


Figure 9 depicts the spectral analysis algorithm through an example PSD of Pitot-derived airspeed (top tile) and coldwire-derived temperature (bottom tile) plotted against sampling frequency. The raw PSD (solid blue) is binned into 9 logarithmically spaced frequency bins between $2 - 200$ Hz and averaged (red dots). The bin-averaged PSD points are then least-squares fit to a $f^{-5/3}$ slope line, omitting spectral artifacts (e.g. 95 and 105 Hz points in the top panel) to determine the mean and standard
deviation of the spectral fit. This is used to provide fit-qualified estimates of $\epsilon$ and $C_T^2$ for each $5$ s time interval.

Additional derived parameters quantifying the stability of the atmosphere are also computed. First, the coldwire temperature, 2D mean wind vector, and calibrated pressure altitude are filtered using a zero-phase distortion digital filter and resampled to 10 Hz. Subsequently, the potential temperature ($\theta$), Brunt-Vaisala frequency ($N$), vertical shear ($dU/dz$), and the gradient Richardson number ($Ri$) are calculated.

Following the criterion described by Muschinski and Wode (1998) and Dalaudier et al. (1994), temperature gradients on the order of $17\Gamma$ ($\Gamma$ - adiabatic lapse rate) were used to identify the edges of stable sheets. A total of 58 individual stable sheet structures roughly 25 to 50 m thick were identified from the DH2 measurements in this campaign. Stability ducts, consisting of large $N^2$ sheets constraining weakly stable and weakly turbulent layers as deep as 400 m, were also prevalent (see $\epsilon$ in Figures 11 and 13). Such structures, often persisting up to five hours under very stable conditions, were commonly observed near the
peak altitude of Granite Mountain (850 to 900 m AGL). Altitude undulations in persisting stable structures (see $N^2$ in Figure 11 and 13) during strong (8 to 10 ms$^{-1}$) eastward-wind forcing over Granite mountain suggest the presence of mountain waves. Temperature gradients as steep as 0.18 Km$^{-1}$ or $\sim 18\Gamma$ (with tropospheric dry adiabatic lapse rate $\Gamma \sim 9.8 \times 10^{-3}$ Km$^{-1}$) were typically observed across most sheets.

A shallow nocturnal boundary layer (200 m deep), with recurring sheet activity at the mountain top (850 m) and higher aloft
(1450 - 1600 m) separated by deep, intermittent turbulent layers (600 - 800 m deep) was the general theme underlying the observations at FS1. Escalation of shear at altitudes coincidental with the undulating sheet pairs may be responsible for their recurring decay, providing intermittent forcing for the turbulence inside the layers.

Figures 10, 11, 12, 13 show altitude profiles of the DH2 measured and estimated parameters for the vertically sounding aircraft (A1 in Figures 6 and 7) from two sorties deployed at FS1 (see Figure 1) on 6 November 2017. The ascent (solid red
line) and descent (solid blue line) data are separated by a fixed offset to more easily visualize the evolution of various flow features. Also depicted in these figures are the profiles obtained from radiosonde deployments at FS2 that were co-ordinated with the DH2 sorties.

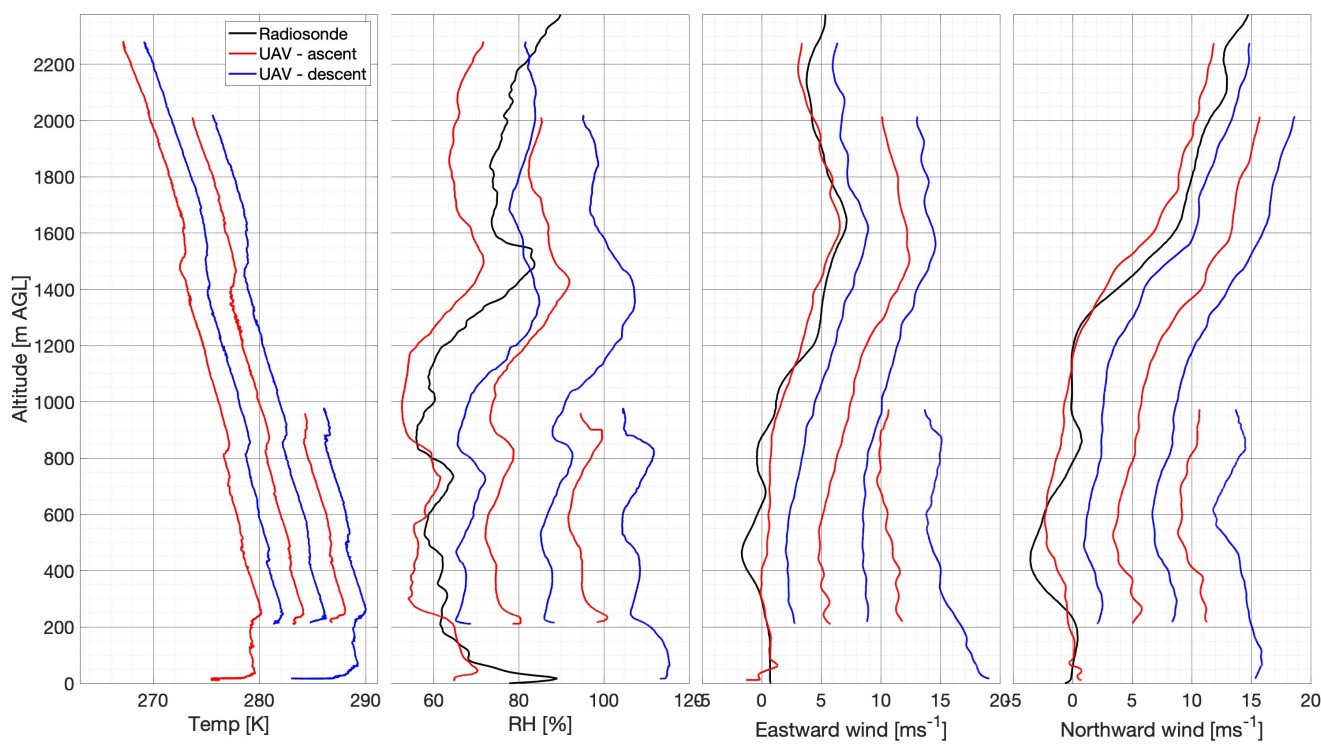

**Figure 10.** Vertical profiles (offset for clarity) of 800 Hz temperature (offset: 2 C), 100 Hz humidity (offset: 10%), 10 Hz horizontal wind speed (offset: 3 ms$^{-1}$) and direction (offset: 180°), eastward and northward winds (offset: 3 ms$^{-1}$) from the vertical sounding aircraft in S1 on 6 November 2017. Red (blue) lines indicate ascending (descending) flight legs. Black lines correspond to radiosonde data.

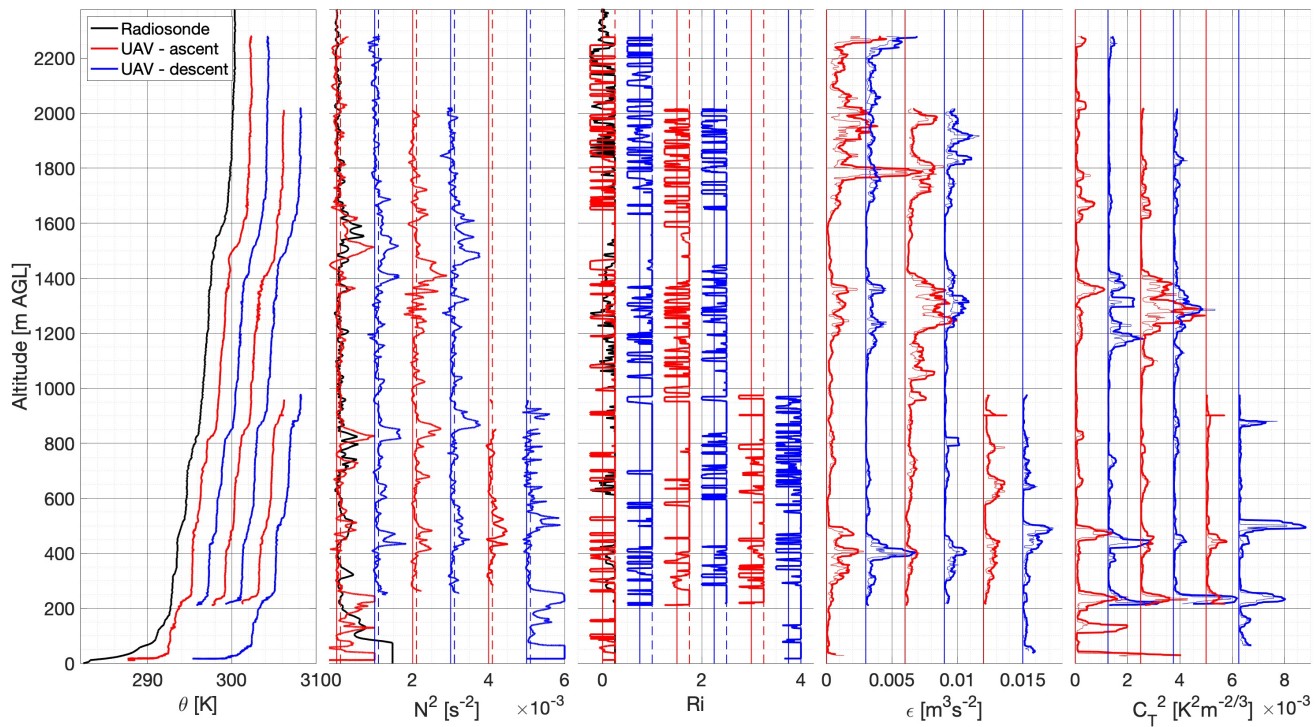

**Figure 11.** Vertical profiles (offset for clarity) of 800 Hz $\theta$ (offset: 2 K), 10 Hz $N^2$ (offset: $10^{-3}$ s$^{-2}$, dash line: $N^2 = 10^{-4}$ s$^{-2}$), 10 Hz $Ri$ (offset: 0.75, dash line: $Ri = 0.25$), $\epsilon$ (offset: $3 \times 10^{-3}$ m$^2$s$^{-3}$) and $C_T^2$ (offset: $1.25 \times 10^{-3}$ C$^2$m$^{-2/3}$) estimated using 5 s time records from the vertical sounding aircraft in S1 on 6 November 2017. The thick lines in the $\epsilon$ and $C_T^2$ tiles show the 30 point moving-average means of the 5s interval estimates (thin lines).

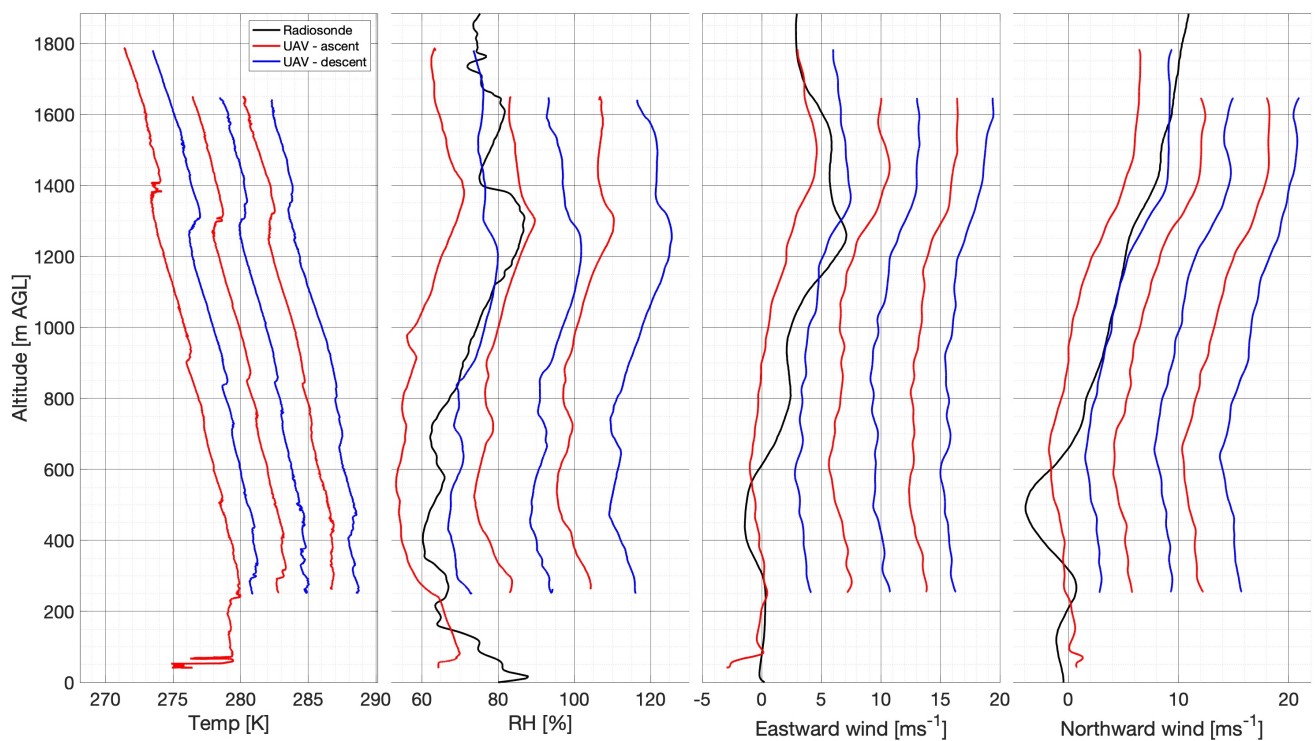

**Figure 12.** Vertical profiles of temperature, relative humidity, horizontal wind speed and direction, Eastward and Northward Winds from the vertically sounding (red ascent, blue descent) aircraft in S2 on 6 November 2017. Radiosonde data are indicated with black lines.

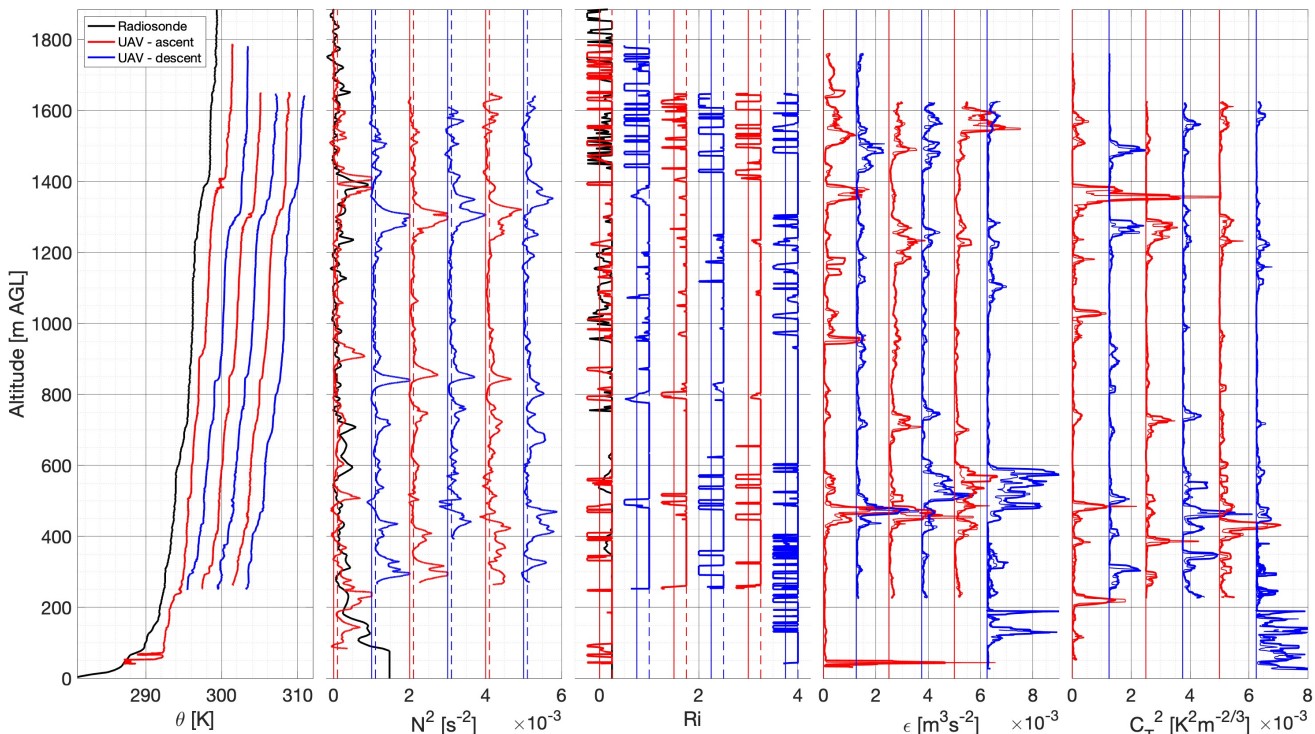

**Figure 13.** Vertical profiles of potential temperature, buoyancy frequency, gradient Richardson number, TKE dissipation rate (offset: $1.25 \times 10^{-3}$ $\mathrm{m^2 s^{-3}}$), and temperature structure function from the vertically sounding aircraft in S2 on 6 November 2017. The thick lines in the $\epsilon$ and $C_T^2$ tiles show the 30 point moving-average means of the 5s interval estimates (thin lines).

The sorties presented in Figures 10, 11, 12 and 13 show a stable nocturnal boundary layer extending to 200 m, capped by an inversion layer. Eastward wind aloft up to $12$ $\mathrm{ms^{-1}}$ was observed (see figures 10 and 12). A strong speed shear developed between 1200 and 1600 m AGL. The background atmospheric column was near-neutrally stable ($N^2 = 10^{-5}s^{-2}$). The DH2 observed an undulating sheet in the measurements of temperature and humidity at 800 m and another sheet formation at 1500 m (see the tile on the left in Figures 10 and 12). An intermittent patch of weak turbulence, 200 m deep, was observed between 1400 and 1600 m (see $\epsilon$ and $C_T^2$ tiles in Figure 11). Subsequent vertical profiles of $\epsilon$ and $C_T^2$ in Figures 11 (from S1 at 03:00 LT) and 13 (from S2 at 05:00 LT) exhibit signs of diminishing turbulence likely leading to re-laminarization, enabling formation of a steep temperature gradient that is characteristic of a highly stable sheet. The extinction of turbulence is apparent in the abrupt reduction of $\epsilon$ just below 1400 m from the first ascent to the first descent in Figure 13. The confined yet elevated levels of $C_T^2$ immediately below 1400 m in Figure 13 further supports this conclusion.

The $\theta$ and RH profiles from the hourly radiosonde soundings monitoring the atmospheric column at FS2 (9.5 km downstream of FS1) on 6 November 2017 (not shown here) suggested that the two stable sheets observed by DH2 at FS1, shown in Figures 10 and 11 (at 800 m and 1300 m), are highly localized and likely dissipate as they advect. The undulating motions exhibited by the sheets imply wave activity in the lee of Granite mountain due to strong eastward wind forcing near the surface. Preliminarily,

this case seems to support the analysis presented in Balsley et al. (2018); Fritts et al. (2013); Fritts and Wang (2013) implying that S&L structures are maintained by GW-FS interactions.

Preliminary analysis of the measurements made by laterally sampling aircraft (A2 and A3 aircraft from each sortie) is underway. The 2D scatter plots of T/RH, $\theta$, Pitot and hotwire derived $\epsilon$, and coldwire derived $C_T^2$ as a function of altitude and longitude for the laterally sampling A3 aircraft from S1 on 6 November 2017 are shown in Figures 14 and 15. The uniform T/RH and $\theta$ along the lateral extent of the sheets (1.5 km) between 1200 and 1400 m in Figures 14 and 15 indicate that this shallow layer exhibits uniform temperature and stratification across the lateral measurement extent suggesting that the layer spans at east 1.5 km kilometers laterally. Additionally, the DH2 derived estimates of $\epsilon$ and $C_T^2$ between 1200 m and 1400 m in Figure 15 support this. An intermittent patch of turbulence approximately between 500 m and 700 m laterally and at an altitude of 1300 m in Figure 15 exhibits increased $\epsilon$ in comparison to its surroundings. Such localized intermittent patches of turbulence with varying $\epsilon$ intensities are typical of early stages of growing turbulence within the layer. Also, the lateral and vertical variability in the estimates of $C_T^2$ inside the layer (see Figure 14) implies ongoing mixing as a consequence of persistent turbulence.

The slanted lateral sampling strategy employed by A2 (and A3) aircraft is expected to assist in decoupling the spatial and temporal variations in the measurements of these unsteady, localized, and advecting turbulence patches and may provide insights on the intermittency and evolution of small-scale weak turbulence within the shallow mixed layers. Although more thorough analysis of DH2 data products is expected to provide further insights on the lateral structure and variability of S&L, it is unlikely to shed much light on the formation mechanism. For this, high resolution DNS modeling may provide more insights, and these DH2 measurements can provide qualitative information on the background structure and forcing conditions invaluable for initializing and validating future DNS studies of such multiscale turbulence dynamics in the free atmosphere.

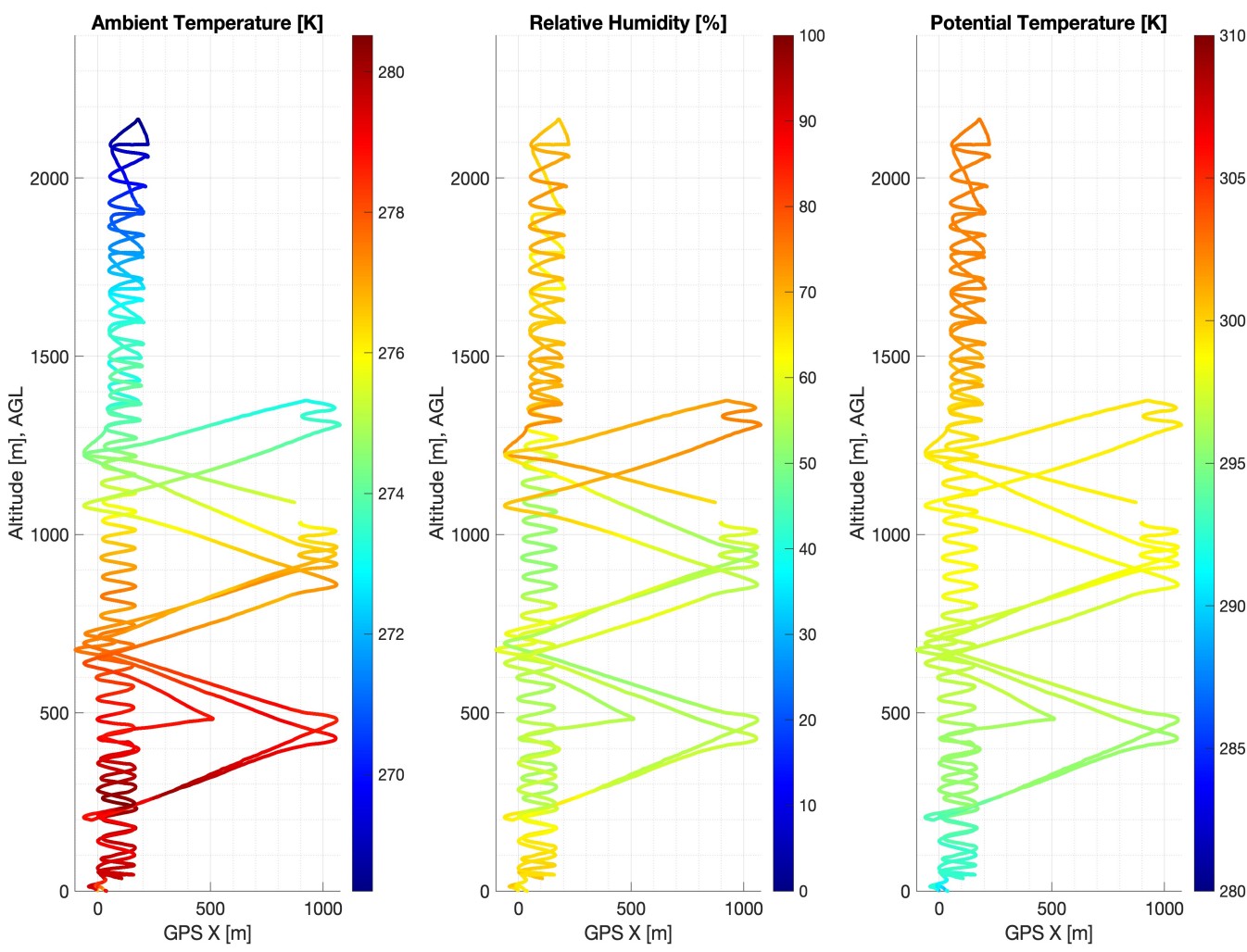

**Figure 14.** Scatter plot of T, RH, and $\theta$ plotted (colors) as a function of altitude and longitude for the lateral sounding DH2 aircraft from the first sortie (S1) on 6 November 2017.

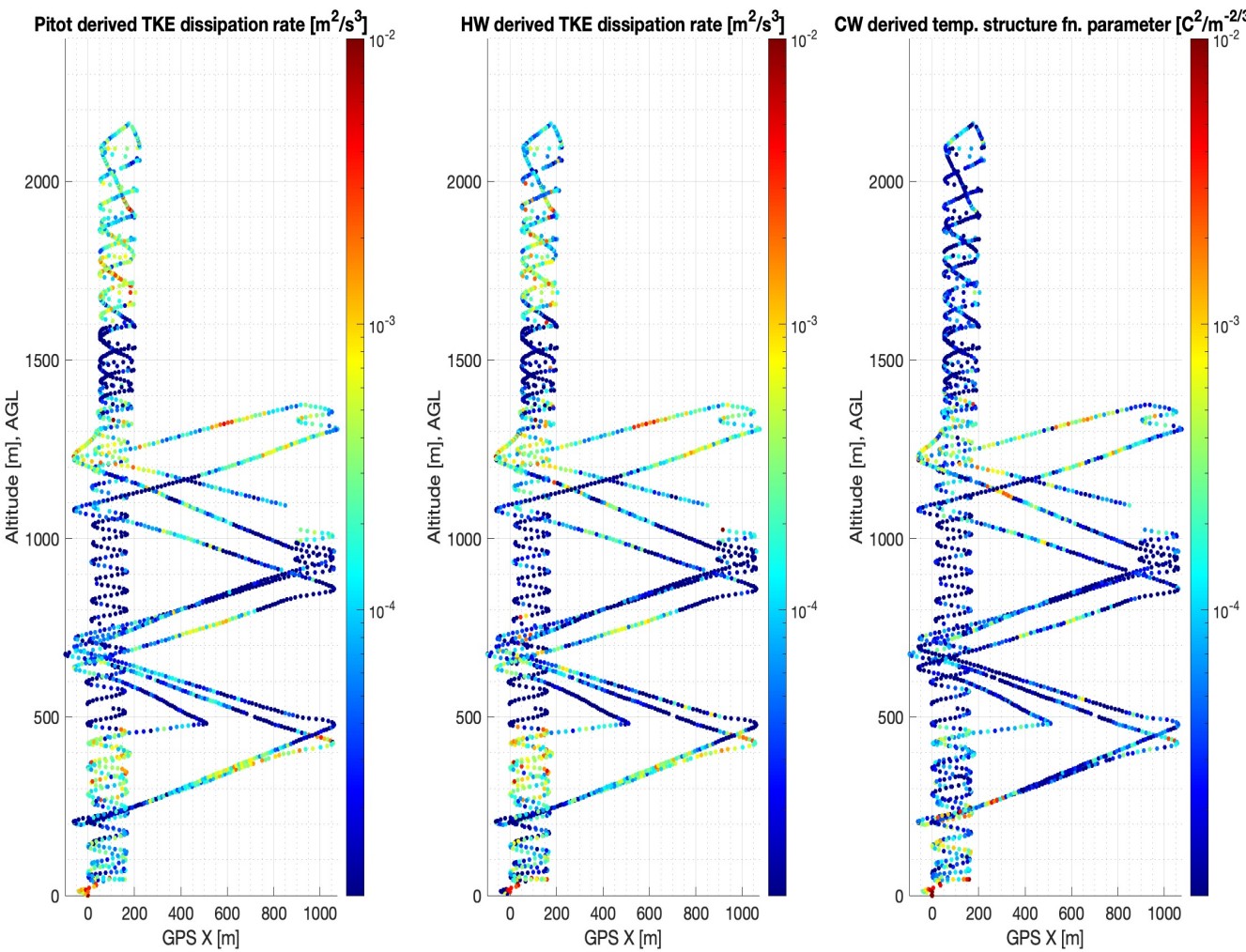

**Figure 15.** Scatter plot of Pitot and hotwire derived $\epsilon$ and coldwire derived $c_T^2$ plotted (colors) as a function of altitude and longitude for the lateral sounding DH2 aircraft from the first sortie (S1) on 6 November 2017.

## 5 Conclusions and Future work

Sheet and Layer (S&L) structures appear to be ubiquitous in the nocturnal boundary layer under relatively quiescent conditions, often extend to higher altitudes in the lower troposphere, and have parallels extending to much higher altitudes at larger spatial scales. The IDEAL program was motivated by multiple previous radar observations and high-resolution in-situ profiling measurements revealing S&L structures, along with initial high-resolution DNS modeling of KHI MSD demonstrating the emergence of S&L dynamics from idealized initial conditions. The resulting field campaign employing high-resolution DH2 UAS instrumentation and the observation strategy employed at DPG in Utah with guidance from daily local WRF weather

forecasts, ISS wind profiler radar and hourly radiosonde profiles provides a unique data set with multi-UAS sorties performing

coincident, but diverse, flight profiles in a common volume under stable nighttime conditions.

The vertical profiling aircraft, deployed first during each sortie, surveyed the atmospheric column and relayed in real-time to the ground station. This information was used to guide and steer the lateral profiling aircraft in each sortie. This sampling strategy increases the diversity of observations of S&L dynamics for characterizing unsteady, advecting, weak and intermittent turbulence layers in the free troposphere (up to 3 km). Preliminary analysis of lateral profiling aircraft measurements identified

a range of behavior, from weak intermittent turbulence to strong persisting turbulent layers, as well as highly localized turbulent layers at 850 m and 1500 m in the leeward side of Granite Mountain. The initial assessment of data from lateral sounding aircraft on 6 November 2017 presented in Section 4 were found to be qualitatively consistent with the description of the S&L structures described by Tjernström et al. (2009) and Balsley et al. (2018). Although preliminary analysis of lateral profiling DH2 UAS are expected to provide valuable insights on the morphology and evolution of shallow turbulent layers, a detailed

analysis and interpretation of this observational data is presently hampered due to the complexity in distinguishing the spatial and temporal variability in the data due to steady, localized and advecting turbulence patches (Wainwright et al., 2015). High-resolution modeling can be instrumental in providing context and phenomenology for interpreting these measurements.

Recently, DNS models designed to further explore KHI MSD explored misaligned KHI billows leading to cases of "tube and knot" (T&K) dynamics. These secondary KHI mechanisms were examined in detail by Fritts et al. (2021a) and Fritts et al.

(2021b). The T&K dynamics resulted in secondary KHI and transitions to turbulence that were dramatically more aggressive and intense than in their absence. The KHI MSD DNS also predicted the emergence of small-scale KHI dynamics within induced S&L structures that emerged from the idealized initial conditions. The expected wide-spread presence of MSD suggest that KHI T&K dynamics in the atmosphere are likely major contributors to the small-scale dynamics and resulting S&L structures. The high-resolution turbulence observations carried out as part of the IDEAL program have characterized multiple

events that are expected to expand such DNS studies in exploring the involvement of KHI MSD in forming and driving S&L structures and accompanying turbulence in the lower atmosphere. Important among these will be additional modeling of KHI T&K dynamics induced by MSD at larger scales. These dynamics may be especially beneficial, given their recent discovery in atmospheric observations and initial DNS modeling results suggesting that they may have important implications for mixing and induced larger-scale motions in the atmosphere, oceans, and other stratified and sheared fluids.

*Data availability.*

The observational data from sUAS, ISS soundings and 915 MHz radar wind profiles, DPG 449 MHz radar wind profiles, SAMS, mini SAMS and ancillary data are available for download on request at https://www.eol.ucar.edu/field_projects/ideal in standard binary formats (.netcdf and .mat). Documents describing processing and quality control for all the platforms along with the metadata files are made available with the data sets. The suite of scripts developed to process and analyze DH2 and

radiosonde measured data during the IDEAL observation program are maintained on a private GitHub repository and will be shared on request by correspondence with the lead author.

*Acknowledgements.* The IDEAL field program and modeling efforts were supported by the US National Science Foundation (NSF) grants AGS-1632772 and AGS-1632829, with related modeling efforts supported by AGS-2032678 and AGS 1758293. The authors would like to acknowledge the efforts of Tyler Mixa and Kam Wan of GATS for their assistance in the performance of DH2 sUAS measurements during
the field program. Isabel Suhr at NCAR EOL assisted William Brown with radiosonde launches. The authors would also like to acknowledge the DPG Meteorology division for providing detailed weather forecasting, reports, and briefings.

*Author contributions.*

DF conceived the joint IDEAL measurement and modeling program and guided the initial DNS modeling components with assistance from TL and LW at GATS. DL developed the DH2 sUAS vehicles and AD assisted in developing and testing the sensing systems. DL led
the DH2 portion of the field campaign and AD assisted in DH2 deployment. AD and DL carried out the preliminary DH2 data analyses. WB led the ISS siting, operations, and data handling activities, and DZ hosted the campaign at DPG, including coordination of daily weather briefings. DF and LK provided (remote) science guidance during the campaign.

*Competing interests.*

The authors have no competing interests regarding this work.

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

## Appendix

Table 4 lists all the DH2 UAS sorties deployed during IDEAL field campaign.

| Sortie number | Date | Flight launch time (HH:MM) - LT | Flight launch site | Flight number | Flt. ceiling and Meas. strategy |
|---|---|---|---|---|---|
| S01 | 24-Oct-17 | 06:08 | FS2 | DH03 DH04 | DH03 - spiral; ceiling - 1000 m DH04 - slant racetrack between 50 - 1100 m |
| S02 | 26-Oct-17 | 04:05 | FS2 | DH05 | DH05 - spiral; ceiling - 1700 m |
| S03 | 26-Oct-17 | 05:53 | FS2 | DH06 DH07 | DH07 - spiral; ceiling - 1300 m DH06 - slant racetrack between 300 - 1100 m |
| S04 | 28-Oct-17 | 02:55 | FS2 | DH08 | DH08 - spiral; ceiling - 1800 m |
| S05 | 28-Oct-17 | 05:12 | FS2 | DH09 DH10 | DH09 - spiral; ceiling - 2100 m DH10 - slant racetrack between 900 - 1100 m |
| S06 | 30-Oct-17 | 02:44 | FS2 | DH11 DH12 DH13 | DH11 - spiral; ceiling - 2200 m DH12, DH13 - slant racetrack between 200 - 2100 m |
| S07 | 31-Oct-17 | 05:07 | FS2 | DH14 DH15 | DH14 - spiral; ceiling - 2550 m DH15 - slant racetrack between 250 - 2550 m |
| S08 | 01-Nov-17 | 03:16 | FS1 | DH16 DH17 | DH17 - spiral; ceiling - 3100 m DH16 - slant racetrack between 200 - 2700 m |
| S09 | 01-Nov-17 | 06:00 | FS1 | DH18 DH19 | DH19 - spiral; ceiling - 1550 m DH18 - slant racetrack between 200 - 1550 m |
| S10 | 02-Nov-17 | 02:50 | FS2 | DH20 DH21 | DH21 - spiral; ceiling - 2700 m DH20 - slant racetrack between 50 - 2700 m |

| Sortie number | Date | Flight launch time (HH:MM) | Flight launch site | Flight number | Meas. strategy and target alt. |
|---|---|---|---|---|---|
| S11 | 02-Nov-17 | 05:12 | FS2 | DH22<br>DH23 | DH22, DH23 - slant racetrack between 50 - 2600 m |
| S12 | 02-Nov-17 | 06:57 | FS2 | DH24 | DH24 - spiral; ceiling - 1600 m |
| S13 | 03-Nov-17 | 02:54 | FS2 | DH25<br>DH26 | DH26 - spiral; ceiling - 3100 m<br>DH25 - slant racetrack between 50 - 3100 m |
| S14 | 03-Nov-17 | 04:48 | FS2 | DH27<br>DH28 | DH28 - spiral; ceiling - 3100 m<br>DH27 - slant racetrack between 50 - 3100 m |
| S15 | 03-Nov-17 | 06:34 | FS2 | DH29<br>DH30 | DH30 - spiral; ceiling - 3000 m<br>DH29 - slant racetrack between 50 - 3000 m |
| S16 | 06-Nov-17 | 02:45 | FS1 | DH31<br>DH32<br>DH33 | DH33 - spiral; ceiling - 2400 m<br>DH31, DH31 - slant racetrack between 200 - 2200 m |
| S17 | 06-Nov-17 | 05:17 | FS1 | DH34<br>DH35<br>DH36 | DH36 - spiral; ceiling - 1800 m<br>DH34, DH35 - slant racetrack between 200 - 1600 m |
| S18 | 07-Nov-17 | 02:44 | FS1 | DH37<br>DH38 | DH38 - spiral; ceiling - 2400 m<br>DH37 - slant racetrack between 900 - 2300 m |
| S19 | 07-Nov-17 | 04:37 | FS1 | DH39<br>DH40 | DH40 - spiral; ceiling - 2000 m<br>DH39 - slant racetrack between 800 - 1500 m |
| S20 | 07-Nov-17 | 06:24 | FS1 | DH41 | DH41 - spiral; ceiling - 2100 m |
| S21 | 09-Nov-17 | 03:02 | FS2 | DH42 | DH42 - spiral; ceiling - 1800 m |
| S22 | 09-Nov-17 | 05:59 | FS2 | DH43<br>DH44<br>DH45 | DH45 - spiral; ceiling - 1600 m<br>DH43, DH44 - slant racetrack between 100 - 600 m |

| Sortie number | Date | Flight launch time (HH:MM) | Flight launch site | Flight number | Meas. strategy and target alt. |
|---|---|---|---|---|---|
| S23 | 10-Nov-17 | 02:53 | FS1 | DH46 DH47 DH48 | DH48 - spiral; ceiling - 2600 m DH46, DH47 - slant racetrack between 200 - 2500 m |
| S24 | 10-Nov-17 | 05:12 | FS1 | DH49 DH50 DH51 | DH51 - spiral; ceiling - 2600 m DH49, DH50 - slant racetrack between 500 - 2500 m |
| S25 | 11-Nov-17 | 02:47 | FS1 | DH52 DH53 DH54 | DH54 - spiral; ceiling - 3000 m DH52, DH53 - slant racetrack between 200 - 2800 m |
| S26 | 11-Nov-17 | 04:42 | FS1 | DH55 DH56 | DH56 - spiral; ceiling - 950 m DH55 - slant racetrack between 150 - 900 m |
| S27 | 13-Nov-17 | 02:53 | FS1 | DH57 DH58 DH59 | DH59 - spiral; ceiling - 3100 m DH57, DH58 - slant racetrack between 200 - 3100 m |
| S28 | 13-Nov-17 | 06:22 | FS1 | DH60 DH61 DH62 | DH62 - spiral; ceiling - 3100 m DH60, DH61 - slant racetrack between 1700 - 2800 m |
| S29 | 14-Nov-17 | 02:53 | FS2 | DH63 DH64 DH65 | DH65 - spiral; ceiling - 2200 m DH63, DH64 - slant racetrack between 100 - 1400 m |
| S30 | 14-Nov-17 | 05:08 | FS2 | DH66 DH67 | DH67 - spiral; ceiling - 2400 m DH66 - slant racetrack between 800 - 1400 m |
| S31 | 15-Nov-17 | 02:44 | FS1 | DH68 DH69 DH70 | DH70 - spiral; ceiling - 3100 m DH68, DH69 - slant racetrack between 1100 - 2400 m |

**Table 4.** List of DH2 UAS sorties deployed during IDEAL observation campaign. The launch date, site, time (HH:MM, LT), flight ceiling and observation strategy for aircraft in each sortie are noted.