# Peer review of "Instabilities, Dynamics, and Energetics accompanying Atmospheric Layering (IDEAL): High-Resolution in-situ Observations and Modeling in and above the Nocturnal Boundary Layer"

_Atmospheric Measurement Techniques, 2021_

## Author Response (AR1)

**Reviewer comments and author responses for the IDEAL overview paper – AMT Journal**

**Reviewer 1 – Comments and responses**

Doddi et al. provide a manuscript about the IDEAL measurement campaign. The IDEAL program and the associated campaign target a very relevant and interesting topic of atmospheric research, which is the structure of the lower troposphere in strongly stable conditions. New ways of sampling stability and turbulence with multiple UAS were explored in the campaign which could contribute significantly to a better understanding of the dynamics under such conditions.

I think the authors are not very clear with the concept of the manuscript. While the title and abstract suggest a focus on the observational campaign, the introduction and section 5 suggest that they want to give an overview of the research program IDEAL as a whole. I think the manuscript should be revised in either of the two directions. If the authors decide to focus on the observations, I request some major revisions as described in the general and specific comments below, before I can recommend the manuscript to be published in AMT. If they decide to describe the whole research program, they might want to consider resubmitting to ACP instead.

Our objective is to present an overview of the IDEAL program (as described in the penultimate paragraph of the Introduction section) with emphasis on the observation phase (phase I) that has already been carried out, and its scope in guiding the DNS during Phase II. The last paragraph of the introduction section states that the article focuses on the field campaign and related topics. For the revision we plan to explicitly state the implications of the measurements in guiding DNS during Phase II. The authors believe this will more clearly keep the focus on the field campaign while still explaining how DNS is proposed to be conducted.

Further, the authors believe that it is common for field campaigns, like IDEAL, to present a paper detailing the campaign, its objectives, and present the dataset outlining the preliminary results before following up with a detailed look at the synthesis and findings. AMT does not preclude such articles. We intend to follow up with papers discussing the data processing and analysis techniques, and significant findings on various issues of scientific relevance addressed by the campaign.

**General comments:**

- The introduction is very much focused on the sheet & layer research. If the focus of the paper should be the description of the observational campaign and in particular the UAS measurement system and flight strategies, there should also be some references to similar

campaign setups and other UAS systems. UAS have been excessively used in boundary-layer research for vertical profiling, but also turbulence measurements and even combinations of both with multiple systems operating simultaneously. The IDEAL campaign should be put into some context, including not only other DataHawk campaigns.

The literature review conducted by the authors has not revealed any field campaigns which focused exclusively on S&L observations (except perhaps the ShUREX2016-2017 campaigns employing the DataHawk UAS; Here, exploring the S&L dynamics was one of the main objectives besides BL observations and turbulence emanating from convection sources in the free-atmosphere). However, numerous authors have reported observations of S&L events from the measurements of radiosondes, instrumented towers, tethered lifting systems (balloons, kites, etc.), VHF and UHF radars, and combinations of these instruments during field campaigns perhaps designed to observe other atmospheric phenomena. Putting this perspective into the context of IDEAL campaign, we feel, is necessary because IDEAL observation program was conceived with the sole purpose of characterizing S&L dynamics and accompanying turbulence.

However, the point that DH2 measurements during IDEAL set out to observe S&L structures exclusively was not made apparent to the reader. The paragraph on line 52 of introduction feels like a natural place in the article to do so.

The authors will include additional literature describing observations of S&L using various instruments including UAS and use that information to put the objectives of the IDEAL observation campaign into perspective in the introduction section.

- Section 2 gives a lot of details about the UAS which reads a bit like a datasheet or advertisement. On the other hand, the dataset is not very well presented. For a description of the campaign, I expect at least a list of days of measurements with corresponding conditions and flight strategies. It is mentioned later that several different flight strategies were performed with aircraft A2 and A3, but it is never presented when and how often they were performed. This would be important to understand the database better.

The DataHawk UAS employs an inhouse developed Autopilot system utilizing custom electronics. The UAS is equipped with a suite of sensors that best meet various measurement demands set by the science goals of different observation campaigns. Since the DataHawk configuration used during IDEAL observation campaign was different (and unique to IDEAL) from its predecessors, the authors deemed it necessary to describe, in detail, the platform's characteristics and capabilities. Similar examples can be found in the literature: Scipion et al. 2016 and Kantha et al. 2017.

Scipión, D. E., Lawrence, D. A., Milla, M. A., Woodman, R. F., Lume, D. A., and Balsley, B. B.: Simultaneous observations of structure function parameter of refractive index using a high-resolution radar and the DataHawk small airborne measurement system, ann-geophys.net,34, 767–780, https://doi.org/10.5194/angeo-34-767-2016, www.ann-geophys.net/34/767/2016/, 2016.

Kantha, L., Lawrence, D., Luce, H., Hashiguchi, H., Tsuda, T., Wilson, R., Mixa, T., and Yabuki, M.: Shigaraki UAV-Radar Experiment (ShUREX): overview of the campaign with some preliminary results, Progress in Earth and Planetary Science, 4, https://doi.org/10.1186/s40645-017-0133-x, 2017.

We concur that the article is deficient in describing the UAS and radiosonde datasets described in the article. A table listing the background conditions, flight location, dates and time of flight, brief description of observed features, and flight strategy for all UAS sorties (and radiosonde deployments) will be included in the revised manuscript.

Line 210 attempted to describe, vaguely, the sampling strategy employed by aircraft A2 and A3. Currently the paragraph starting on line 209 informs the readers as to why A2 and A3 were deployed but fails to describe how often the lateral sampling was carried out.

Lateral sampling using one or two aircraft was carried out during each UAS sortie. Thus, every UAS sortie consists of two or three aircraft including a vertical sounding aircraft and one or two lateral sounding aircraft.

Including the previously described table (containing UAS dataset information assorted by sorties) would therefore aid in describing the flight operations and the dataset.

- The figures in the manuscript look a lot like copy and pasted from quick looks of the individual instruments. Labels are often small, much information is included that is not described in the caption. I think the authors can do better to prepare them adequately well for a publication. There is also no consistent nomenclature. Examples are zonal and meridional wind vs. eastward and northward wind. Figure labelling is sometimes wind direction, sometimes w_dir. Although the latter are minor issues, they make the manuscript hard to read.

Figures 3, 4, 5, and 7 are quick look plots relayed by different teams (radar, radiosonde, etc.) to the UAS team in real-time. The authors will use quality-controlled datasets (unlike the raw data which is presented in these figures) to recreate publication quality images to replace all these figures in the revised manuscript.

- It appears to me that all the examples of measurement data are from 6 November 2017, but they are mostly presented separately without connecting them. It would be nice to maybe give an introduction to the conditions on this day which serves as a case study and then lead the reader through the findings from different instruments.

Figure 3 presents the SNR, vertical velocity and wind information measured by the VHF radar on the 1st of November. This figure intended to show a possible Kelvin-Helmholtz Instability (KHI) event observed during the campaign.

To better connect sections 2.2 and 2.3 to the following section (section 3), figure 3 will be updated with VHF radar SNR, vertical velocity, and wind from the 6th of November.

- The section about DNS is very much detached from the rest of the manuscript. If the authors decide to focus on the observational campaign, I think the section is not really necessary. If they decide to present the whole project, including the simulations, they should better connect the goals or the findings of the campaign to the presented simulations. I think this is not done very well.

Section 5 presents results from two previously conducted DNS to study the formation of S&L structures arising from superpositions of convectively stable gravity waves (GW) and dynamically stable mean shears. The first DNS experiment featured a GW of amplitude 0.5 (relative to vertical gradient of local potential i.e., $\frac{(d\theta/dz)_{min}}{(d\theta/dz)} = 0.5$) and an intrinsic frequency of N/10 (where N is the Brunt Vaisala Frequency) at Reynolds Number of 50,000. The second DNS experiment designed to study the Kelvin Helmholtz Instability (KHI) assumed a Reynolds number of 5000 and a minimum Richardson Number of 0.1 with a random white noise background velocity field (superimposed to stimulate instability growth leading to KH billows). Figures 17 and 18 present relevant results from these two DNS.

These DNS of multi-scale dynamics (MSD) suggested that the resulting KHI tubes and knots (T&K) dynamics are likely major contributors to the S&L structures which ubiquitously occur in the atmosphere. Thus, these DNS studies presented in section 5 provided the motivations for the IDEAL observation program. In phase II of the IDEAL project, we plan to expand such DNS studies to explore the implications of IDEAL measurements.

This section, as the reviewer points out, is misplaced. It serves our purpose better to present the motivations provided by DNS upfront – within the introduction section. We intend to restructure the implications of these initial DNS studies and present them as motivations within the introduction section.

- The conclusions are very brief and vague. Are there any lessons-learned from the campaign? What were the highlights? What can be done with the dataset as a whole, not only with slanted UAS flights?

Section 3 intends to present preliminary findings from the field campaign as detailed analysis of the UAS datasets are still underway. The article serves the purpose of describing the UAS dataset and the conditions during which the observations were made. The authors will first work on improving the findings presented in section 3 and use that information to discuss and draw conclusions in the final section.

Specific comments:

p.2, l.47: I am not sure what is meant by "dexterity" of measurement platforms.

We intend to convey that UAS provide flexible (dexterous) sampling strategies, both vertically and laterally, when compared to other in-situ measurement platforms (radiosondes, instrumented towers, etc.).

p.2, l.48f: I agree that spatial information could yield many new insights beyond single-point vertical profiles. However, the only data that are presented later in Section 4 are such vertical profiles.

Preliminary analysis was conducted exclusively on vertical sounding aircraft from each UAS sortie mainly due to the familiarity in analyzing and interpreting the vertical sampling aircraft data as this is a typical flight strategy used by fixed-wing aircraft.

Analysis and interpretation of observation data from lateral sounding aircraft is complicated by horizontal winds which laterally advect the S&L structures being observed. Consequently, it is difficult to draw concrete conclusions about the lateral structure of S&L from only preliminary results.

However, the manuscript will be revised to contain 2D scatter plots from one lateral flight showing T/RH, wind components, potential temperature, TKE dissipation rate, and temperature structure function parameter plotted as functions of Latitude vs Longitude, Altitude vs Longitude and Longitude vs Time (lateral measurements were made along 1Km legs roughly aligned East-West). These three figures will help to highlight the spatial structure and temporal evolution (if evident) of the underlying S&L structures from the lateral surveys, as an example of the type of analysis that we (and others) may wish to conduct on this dataset.

p.3, l.58: 72 flights in which time frame? Does this mean single flights, or flights with three UAS in parallel?

The term 'flight' is used for an individual aircraft flight and the term 'sortie' has been used for a coordinated set of flights deployed simultaneously. In this context, we refer to the total number of UAS flights carried out during the IDEAL observation program.

Figure 1: If an elevation map is available, it would be really nice to show the site with contour lines, or color-coded elevation.

The addition of elevation map would certainly be beneficial. We will update figure 1 to show elevation contour lines.

p.5, l.86: "Unbreakable wing trailing edges" - unbreakable seems a bit unrealistic.

"Unbreakable" will be replaced with "resilient" in the revised manuscript.

p.5, l.97: when wind speed exceeds airspeed, the aircraft moves backwards with respect to the ground. What does it mean that the flight is stabilized in that case? If airspeed is controlled, there is in general no flight stability issue, but maybe an issue with navigation.

When course heading is controlled, this reverses sign as the wind exceeds the airspeed, causing the aircraft control to command a 180 degree change in heading, destabilizing the flight. Instead, on the DH2 compass heading is the controlled variable, and this is computed from desired course heading and wind estimates using the wind triangle to produce a "wind-aware" guidance law that causes robust tracking of a desired GPS course in high wind.

p.5, l.98: Synoptic wind means geostrophic wind? What is meant by "aloft"? And why does it limit the ceiling to 3 km exactly?

Yes, in this context we mean geostrophic winds. We see why this creates confusion because it is vague. Instead, the sentence should say, "synoptic winds above 3000m [AGL] typically exceeded 20m/s which limited the flight ceiling to this altitude."

p.6, l.116ff: Is there a reference for this procedure?

The calibration is a simple linear least squares regression between CW measured voltage and the collocated SHT temperature measurements, and as such there is no specific reference for this procedure. This point will be explicitly mentioned in the revised manuscript.

p.7, l.124ff: As above, a reference that describes the sensor fusion and turbulence measurement would be great.
p.7, l.128f: Is there a reference to the wind algorithm? With GPS and airspeed only, the  wind can typically not be retrieved.

The procedures employed to compute turbulence parameters of TKE dissipation rate and temperature structure function parameter, and the estimates of horizontal wind vector components are novel. These estimation algorithms were developed as part of the lead author's (Abhiram Doddi) doctoral thesis and are yet unpublished. The authors are

currently working to describe turbulence and wind estimation procedures in upcoming research articles. References to the lead author's thesis will be made as necessary.

Lawrence and Balsley 2013, Luce 2019 (citations given below) describe the general framework of the estimation procedures utilized for computing wind and turbulence parameters presented in this article.

Table 2:

 - How can the resolution of the vector wind be 0.001 m/s, if airspeed can only be resolved at 0.05 m/s?

This resolution of vector wind should be 0.05 m/s and not 0.001 m/s.

 - unit for dissipation rate and structure parameter accuracy missing. What does the range-value mean in this case?

TKE dissipation rate is in [$m^2/s^3$] and CT2 is in [K^2 m^{-2/3}]. Range is the high end of  of these parameters than can be measured with the instrument that avoids noise floor and power supply limitations.

 - degree symbol missing for all temperature units.

The revised manuscript will be updated to include these.

Figure 3: It would be great if a better quality of the figures could be provided.

The quick look plots in figures 3, 4, 5, and 7 will be replaced with publication quality images using quality-controlled data.

Figs. 4&5: I do not think these figures are really necessary if they are only there to illustrate data that is presented at weather briefings.

The data presented in figure 3 is from the ISS operated VHF radar (915 MHz wind profiler). This information was relayed to the UAS team (in 1-hour installments) periodically during flight operations (between 2:00 – 8:00 am LT).

The weather briefings consisted of observation data from the 445 MHz radar located ~20 Km away from both deployment sites (shown in Figure 1 of the manuscript). The observation data shown in figure 5 was presented to the UAS team at the weather briefings along with the WRF simulation forecasts is shown in figure 4.

Collectively, figures 4 and 5 present the data available to the UAS team before flight deployment on each operational day. The information on synoptic scale flows forecasted in figures 4 and the real-time measurements shown in figure 5 was critical to decide the flight location and flight ceiling, and its inclusion here is meant to accurately describe the operational aspects of the campaign.

p.10, l.170f: The top right panel of Figure 6 does not show wind speeds. It cannot be read from the figure where the first week starts and ends.

The text here should be changed to 'bottom tile'.

Figure 6: I think it is a bit irritating that the x-axis shows sequential soundings and - to my understanding - does not give any information about the time of these soundings. It should at least be clearly indicated which soundings are released in close succession and where there are larger time gaps. It looks as if the plots even feature some interpolation between the profiles, which does not make much sense if the time spacing is not equidistant. I also do not understand why the colormap range is so large for temperature and wind speed.

These figures will be replotted with convenient colormaps and show dates on the X-axis instead of the sounding number. Data interpolation, if used, will be described as necessary.

p.12, l.178f: Where can the stability be seen in the plots and where the intermittent turbulence and sheet structures?

This statement was made based on the plots presented in figures 12 & 14 and should contain a reference to these figures and not figure 3. Therefore, reference to figures 12 and 14 will be made at the end of this statement.

p.13, l.191: I assume "Granite Peak" equals "Granite Mountain"?

Will be changed to read 'Granite Mountain' consistently throughout the manuscript.

Figure 7, caption: I do not see a hodograph as is written in the caption.

Text referring to the hodograph will be removed. Figure 7 is a quick look plot provided. This will be replotted along with figures 3, 4 and 5 to provide publication quality figures using quality-controlled data.

p.14, l.207: In Figures 8,9,10 it looks like the UAS are ascending/descending continuously during the racetrack patterns, but in the text it sounds like they were supposed to stay at dedicated heights. What is correct? Probably what is shown in the figure, but in that case, I do not fully understand the strategy.

The objective of A2 and A3 aircraft was to fly racetracks while continuously ascending and descending within a narrow altitude range (the depth of turbulent layers ~200-500m). But for the flights on November $6^{th}$ (the two flights presented in the text), we decided to fly racetracks in the entire altitude range as the two sheets were separated. This distinction will be made apparent to the reader in the revised manuscript.

p.16, l.210: Ok, so now it is mentioned that the flight strategies for A2 and A3 vary significantly. This was not so clear before and should maybe be mentioned at the beginning of the section.

The flight strategy presented in figures 8, 9 and 10 likely caused this confusion. The description of flight strategy presented in this figure will be presented upfront for clarity.

p.17, l.219: "DH2 identified": How were the stable sheet structures identified? What are the criteria, how is the data processed. A description of this is missing.

"DH2 identified" is ambiguous. This text will be replaced with "58 individual stable sheets and Layers structures were identified…. from the DH2 measured high-resolution CW data".

The criterion we used to identify a sheet was the same as in Muschinski et al 1998. The text will include, briefly, this criterion with reference to the appropriate article.

p.17, l.223: "Altitude undulations": In the flight path? Is this shown somewhere? Is it reflected in wind or temperature measurements as well?

The waterfall plots in figures 12 and 14 show multiple ascent and descent flight legs. The second tile in Figures 12 and 14 (showing $N^2$) suggest that the observed stable sheets (elevated N2 regions at 800m and 1500m in Figure 12; ~800m and 1500m in Figure 14) undulate. It is this undulating motion that we are referring to in this sentence. This

information in apparent from the temperature, and the potential temperature measurements but not the wind measurements.

p.18, l.225: I think this enlarged inset in Figure 12 is not very conclusive. What is this supposed to show?

The inset was presented to suggest that the high-resolution measurements of temperature (800Hz CW temperature) reveal very thin, highly stable sheets that are not often measured by other in-situ instruments.

p.18, l.228: In my opinion the nighttime inversion layer only extends to approximately 100 m. Interestingly, the UAS and radiosonde measurements differ quite significantly in this area. This should be discussed.

The NBL can be very shallow as the reviewer points out and as such it is unlikely to be horizontally homogeneous, especially on scales of tens of kilometers given the terrain near and around Granite Mountain. Radiosondes were launched at a site ~10 km east from the site of UAV deployment and therefore, differences are to be expected. The emphasis in the campaign is not just the NBL but also the free stable atmosphere above the NBL. Radiosondes are to provide additional information about the state of the free atmosphere.

p.18, l.230: Unit missing for N^2

This will be fixed in the revised manuscript.

p.18, l.231f: I do not see from these plots, where undulating temperature and humidity is observed at 800m and especially 1300m.

Same explanation for the undulations as in comment "p.17, l.223".

p.23, l.248: Why is an analysis of A2 and A3 flights not presented? I think this is the essential and new part of the experiment, right?

Yes, we agree that we need to present preliminary analysis plots for this flight strategy. We will include 2D scatter plots from one lateral flight showing T/RH, wind components, potential temperature, TKE dissipation rate, and temperature structure function parameter plotted as functions of Latitude vs Longitude, Altitude vs Longitude and Longitude vs Time (NOTE: The lateral measurements were made along 1Km legs roughly aligned East-West).

Figures 11-14: The time evolution of the profiles is not really discussed. I wonder if it would not be easier to read the plots if only single profiles were shown.

Plots showing timeseries of measured parameters are useful and an example plot for T/RH, winds, potential temperature, TKE dissipation rate, and temperature structure function parameter will be included for one of the A2 (or A3) flights in the revised manuscript.

p.24, l.253f: What DNS code is employed? I realize the references, but if the simulations are introduced here, it would be good to give some basic information.

p.25, l.262: "expanded such MSD studies are contributing" - something seems wrong here.

Figure 17: It would be good to explain somewhere, why the plots are tilted.

This section on DNS, as the reviewer points out, is misplaced (see comment on DNS above). In light of this recommendation, the authors will restructure the implications of these initial DNS studies and present them as motivations within the introduction section and discard the dedicated section on DNS (section 5).

**Reviewer 2 – Comments and responses**

**Overview**

The article by Doddi et al. presents an overview of a project named IDEAL (Instabilities, Dynamics and Energetics accompanying Atmospheric Layering) aiming to achieve a better understanding of the vertical structure of the troposphere under very stable conditions. The project relies on observations and direct numerical modeling (DSN) tools. The observations consist of high temporal resolution measurements acquired from small instrumented unmanned aircraft systems (UAS), Doppler radar profiles, radio soundings, and of meteorological measurements near the surface. A measurement campaign took place in October-November 2017, during which 72 flights of UASs took place, with these flights grouped in pairs or threes. Preliminary results of the field campaign are showcased. Two numerical simulations of Kelvin-Helmholtz instabilities development are also presented.

The IDEAL project is undoubtedly a very interesting atmospheric research topic. The instrumental means implemented on the IDEAL project are relevant and original (in particular the use of fast sensors on guided UAS). However the paper suffers from

some shortcomings, particularly in the description of the data analysis methods. Also, the articulation between observations and modeling, although very interesting in itself, is not very well presented, and I think this aspect should be addressed with more precision.

I therefore recommend that this article be published with some modifications, some minor, others more substantial.

**Major comments**

1) The introductory section (first section) is clear and concise. However, the notion of sheets and layers (S&L) in the present context is not completely clear to me. Does it refer to the alternation of stable and turbulent layers? Or is it strictly limited to the presence of "thin strongly stable non-turbulent" at the edge of weakly stratified layers (presumably turbulent)? It seems to me that the works presented in the second paragraph of the introduction sometimes fall into the first category, sometimes into the second. Can you clarify this S&L notion in the present context? Isn't it necessary to precisely define a sheet (threshold gradients, thickness, location)?

The stable atmospheric column consists of deep homogeneous, but perhaps weakly turbulent layers (where gradients of various properties are negligible) bounded by relatively thin "sheets" with sharp gradients of temperature and humidity. Hence, the "Sheets and Layers" terminology. Such structures are ubiquitous in Doppler Wind profiler radar images and radiosonde soundings. This will be clarified in the revised paper.

2) Several studies of the stable boundary layers partly based on UAS (not only DataHawk) are already published. Also, some results on the properties of turbulent layers in the troposphere have been obtained by careful application of the Thorpe analysis applied to radiosoudings. I think these works should be mentioned in the introductory part.

Few contemporary UAS platforms can provide reliable measurements of TKE dissipation rates and temperature structure function parameter from weak, small-scale turbulence events. Some notable works include Van der Kroonenberg et al 2008, Wildmann et al 2014, Altstadter et al 2015, Baserud et al 2016.

We concur with the reviewer's comment that Thorpe analysis is a reliable technique to infer turbulence characteristics from radiosonde data. A few notable works include Clayson and Kantha 2008, Gong and Geller 2010, Wilson et al 2011, and Kohma et al 2019. However, the pioneering work on the application of Thorpe analysis to radiosonde observation data (Clayson and Kantha 2008) in studying turbulent layers in

the troposphere is acknowledged. The authors will include the above-mentioned studies to strengthen the literature presented in the introductory section. Please note Dr. L Kantha is a coauthor of this paper.

Reference: Clayson, C. A. and L. Kantha, 2008. Turbulence and mixing in the free atmosphere inferred from high-resolution soundings, J. Atmos. Oceanic Tech., 25, 833-852.

Wildmann, N., Ravi, S., and Bange, J. 2014. Towards higher accuracy andbetter frequency response with standard multi-hole probes in turbulencemeasurement with remotely piloted aircraft (RPA).Atmospheric Measure-ment Techniques, 7(4):1027–1041.

van den Kroonenberg, A., Martin, T., Buschmann, M., Bange, J., andV̈orsmann, P. 2008. Measuring the Wind Vector Using the AutonomousMini Aerial Vehicle M2AV.Journal of Atmospheric and Oceanic Technol-ogy, 25(11):1969–1982.

B̈aserud, L., Reuder, J., Jonassen, M. O., Kral, S. T., Paskyabi, M. B., andLothon, M. 2016. Proof of concept for turbulence measurements with theRPAS SUMO during the BLLAST campaign.Atmospheric MeasurementTechniques, 9(10):4901–4913.

Altsẗadter, B., Platis, A., Wehner, B., Scholtz, A., Wildmann, N., Hermann,M., K̈athner, R., Baars, H., Bange, J., and Lampert, A. 2015. ALAD-INA – an unmanned research aircraft for observing vertical and horizontaldistributions of ultrafine particles within the atmospheric boundary layer.Atmos. Meas. Tech., 8(4):1627–1639.

Wilson, R., F. Dalaudier, and H. Luce, 2011: Can one detect small-scale turbulence from standard meteorological radiosondes? Atmos. Meas. Tech, 4, 795-804, doi:10.5194/amt- 4-795-2011.

Gong, J., and M. A. Geller, 2010: Vertical fluctuation energy in US high vertical resolution radiosonde data as an indicator of convective gravity wave sources. J. Geophys. Res., 115, D11110, doi:10.1029/2009JD012265.

Kohma, M., K. Sato, Y. Tomikawa, K. Nishimura, and T. Sato, 2019: Estimate of Turbulent Energy Dissipation Rate From the VHF Radar and Radiosonde Observations in the Antarctic. J. Geophys. Res., 124, doi.org/10.1029/2018JD029521.

**3) Table 2: how are estimated the accuracy of coldwire T? hotwire velocity? What are the characteristics of the instrumental noise on T? and airspeed? (white noise? Noise level? Impact of motor vibration you mentioned?).**

Coldwire temperature (sampled at 800 Hz) is calibrated against a commercial sensor (slow – 100 Hz), so this retains the accuracy specified for this reference sensor. Similarly, hotwire velocity is calibrated against the Pitot-static sensor. In turn, the Pitot-static airspeed is calibrated against GPS speed over each loiter circle as the average of maximum and minimum ground speed.

The turbulence parameters like the TKE dissipation rate and the temperature structure function parameter are estimated by employing spectral analysis of high-cadence CW temperature, HW and pitot airspeed measurements.

Motor vibrations produce periodic artifacts (sharp peaks at specific frequencies) in the HW and pitot airspeed spectra that are excluded in the spectral fitting procedure when estimating the turbulence parameters by an iterative technique that is beyond the scope of this paper. Also, care is taken during spectral analysis to exclude the data close to the sensors' (white) noise floor.

Several (~100 samples) 'quiet' (non-turbulent) spectral samples (calculated using 1s time series of 800 Hz data) were analyzed from CW temperature and pitot airspeed measurements to determine the sensor noise floor. The CW sensor noise floor was estimated to be $1.25 \times 10^{-8}$ $K^2$/Hz. The pitot and HW noise floor was estimated to be at $1.5 \times 10^{-7}$ $m^2/s^2$/Hz.

4) The characteristics of the UASs are described in great detail in section 2. However, almost nothing is said about the data analysis methods.

The authors agree that this is a shortcoming. The procedures employed to compute turbulence parameters of TKE dissipation rate and temperature structure function parameter, and the estimates of horizontal wind vector components are novel. These estimation algorithms were developed as part of the lead author's (Abhiram Doddi) doctoral thesis and are yet unpublished. The authors are currently working to describe turbulence and wind estimation procedures in upcoming and follow-on research articles. Reference will however be made to lead author's thesis.

However, Lawrence and Balsley 2013, Luce 2019 (citations given below) describe the general framework of the estimation procedures utilized for computing wind and turbulence parameters presented in this article. We will include a subsection (in section 2) that briefly describes the estimation procedures.

- With what vertical resolution are the vertical gradients estimated? And why this choice?

The vertical gradients of winds are estimated using pressure altitude data (sampled at 800 Hz) which is filtered and subsampled to 10Hz. This is used to calculate the vertical gradient of horizontal winds. The potential temperature is calculated from the measurements of CW temperature (at 800 Hz), but the buoyancy frequency, and Richardson numbers are calculated using subsampled data (just as above) at 10Hz. For

our preliminary analysis, this resolution provided a reasonable compromise between vertical resolution and overly noisy estimates. Other studies using this dataset may make different choices.

- No estimates of uncertainties on N2, Ri, CT2, epsilon are presented. Can you estimate an error bar for these quantities? Or at least show the dispersion of the estimates?

Uncertainties in estimating epsilon and $C_T^2$ are best described by the variance in the spectral fits to the measured spectra. This information was omitted in the figures presenting these quantities. The manuscript will be revised to include the uncertainties for epsilon and $C_T^2$.

It is not common practice to present uncertainties for N2 and Ri estimates. Instead, the distributions of these quantities are typically found in the literature. We intend to do so in the revision.

- How are turbulent and non-turbulent regions discriminated? (since CT2 and epsilon estimations are meaningless in a non-turbulent region).

Our method of estimating epsilon and $C_T^2$ result in these parameters quantified at every data analysis interval (altitude or time).

In deriving epsilon and $C_T^2$, the measured power spectral density (PSD) calculated over a short interval of time is fit against a model Kolmogorov spectrum (e.g., Tatarskii 1961, Frehlich et al 2003). In case of the measurements obtained from sampling in non-turbulent regions, the measured PSD exhibit very poor fits to the model Kolmogorov spectra (do not exhibit an f^-5/3 slope in PSD vs f). This results in large fit errors, enabling these intervals to be excluded from subsequent analyses, or flagged for more detailed scrutiny of the spectral data, as needed for the analysis at hand.

- The profiles of figures 11-16, from DH2 or radiosondes appears very smooth. Are they filtered? If so, with which filter? And why did you choose these filtering characteristics?

We concur that a brief description of the estimation procedures for the parameters presented in these figures is warranted. For instance, the resolution of turbulence parameters depends on the time series intervals employed during spectral analysis (here, it is 1Hz because 1s intervals have been used to estimate epsilon and $C_T^2$), and the resolution of wind estimates depends on the interval over which the estimates are averaged (here, it is 1Hz because averaging is conducted over a duration of 1s).

We will include a subsection (in section 2) to briefly describe the procedures used to estimate wind and turbulence parameter in addition to potential temperature, $N^2$, and gradient Ri.

5) Figures 5 and 7 are not very useful to describe the strategy of the observations, the description is sufficient. On the other hand, I think that one or two figures showing the power spectral density of T and airspeed to illustrate the estimation method of CT2 and epsilon would have been relevant in the present paper.

As described in the previous comment, a subsection explaining (briefly) the estimation procedures will be included with spectra each for CW temperature and pitot (and HW) airspeed.

6) The link between the fifth part (modeling) and the rest of the paper is not very clear. Was the choice of parameters for the simulations (characteristics of the gravity wave, the tube and the nodes) guided by the observations previously shown?

Section 5 presents results from two previously conducted DNS to study the formation of S&L structures arising from superpositions of convectively stable gravity waves (GW) and dynamically stable mean shears. The first DNS experiment featured a GW of amplitude 0.5 (relative to vertical gradient of local potential i.e., $\frac{(d\theta/dz)_{min}}{(d\theta/dz)} = 0.5$) and an intrinsic frequency of N/10 (where N is the Brunt Vaisala Frequency) at Reynolds Number of 50,000. The second DNS experiment designed to study the Kelvin Helmholtz Instability (KHI) assumed a Reynolds number of 5000 and a minimum Richardson Number of 0.1 with a random white noise background velocity field (superimposed to stimulate instability growth leading to KH billows). Figures 17 and 18 present relevant results from these two DNS.

These DNS of multi-scale dynamics (MSD) suggested that the resulting KHI tubes and knots (T&K) dynamics are likely major contributors to the S&L structures which ubiquitously occur in the atmosphere. Thus, these DNS studies presented in section 5 provided the motivations for the IDEAL observation program. In phase II of the IDEAL project, we plan to expand such DNS studies to explore the implications of IDEAL measurements.

This section, as the reviewer points out, is misplaced. It serves better purpose to present the motivations provided by DNS upfront – within the introduction section. We will restructure the implications of these initial DNS studies and present them as motivations within the introduction section.

**Specific comments**

Granite Peak (in text)→ Granite Mountain (in figures): please, use the same notations throughout.

This will be changed to "Granite Mountain" everywhere in the revised manuscript for consistency.

Line 171: top right panel of Figure 6 shows RH, not surface winds

Yes, we will make the needed change in the revised manuscript.

Figure 6, and line 171: wind "from the South" are negative (lower left panel of Fig.6). Is this correct?

Yes, southerly winds are negative. However, we realized that the range on this figure is not helpful. We will replot this figure with a colormap showing better data range for clarity.

Figure 6: the x-axis should show the dates of the soundings rather than their numbers. Also, the profiles should be visualized according to their dates, thus avoiding interpolating between soundings from one night to the next (which makes no sense).

This detail was also highlighted by other reviewers. In the revised manuscript, this figure will be replotted with the dates of soundings on the X-axis while omitting data interpolation.

Line 219: you mention 31 multi-aircraft sorties. But in line 109, you mention 14 + 13 sorties. Where does the difference come from?

A total of 31 sorties were carried out of which data from 27 (14+13) sorties were processed and analyzed as the other 4 sorties (consisting of 6 flights (1+2+2+1) in total) contained corrupt data. Therefore, these datasets were discarded. This detail is not mentioned in the manuscript. We intend to include table listing the background conditions, flight location, dates and time of flight, brief description of observed features, and flight strategy for all UAS sorties.

Line 230: "The background atmospheric column was near-neutrally stable..." Where, and when? (I don't really see this in either Figure 12 or Figure 14)

The background N2 value (away from the turbulent layers and sheets) averaged to $10^{-4}$ $s^{-2}$. The dashed red and blue vertical lines in the N2 tiles of figures 12 and 14 represent

the $10^{-4}$ $s^{-2}$ values for each profile. We refer to this miniscule N2 value are near-neutrally stable.

Line 239: I do not see any sheet at 1300 m on Figure 11 or 12. Do you mean 800 m on Figure 12 and 1300 m on Figure 14?

Line 239 was meant to comment on figure 14. Not figures 11 and 12. The figure reference will be fixed in the revised manuscript.

Line 240: "The oscillating motion exhibited by the sheets..." What evidence of an oscillation?

The mean height of the sheets (at 800 m and 1300 m) identified from the N2 profiles in figure 14 undulates. We infer from this detail that the sheets are oscillating/ undulating.

---

## Referee Report (RR1)

Review report on the revised version of the paper **"Instabilities, Dynamics, and Energetics accompanying Atmospheric Layering (IDEAL) Campaign: High-Resolution in situ Observations above the Nocturnal Boundary Layer"** by Doddi et al. submitted to the journal Amospheric Measurement Techniques.

**Overview**

The authors have answered most of my questions and objections satisfactorily. The structuring of the article is much improved, as is the quality of the figures. Useful clarifications have been added. The list of references has also been completed.

However, I still regret the absence of an evaluation of the impact of instrumental noise. Except for the position of the fitting line on the PSDs, no error bar is indicated, either on the measurements (T, velocities), or on the deduced quantities (theta, N2, Ri, epsilon, CT2). Taking into account the instrumental noise would certainly be a plus for such a paper presenting data and analysis methods. About the only error bar shown, on the fit of the -5/3 slope line on the PSDs of T and v, how is it estimated? (it is not an uncertainty on the slope of the line since it is fixed a priori).

**Conclusion**: in view of the improvements made to the manuscript, I consider that it can be published in AMT. I recommend, however, to include a consideration of instrumental noise, either by specifying the uncertainties on the measurements and the inferred quantities (could be done in the text), or by explaining why it is not possible to estimate them.

---

## Author Response (AR2)

**Reviewer comments and author responses for the IDEAL overview paper – AMT Journal**

**Reviewer 1: Comments**

Overview:
Doddi et al. provide a manuscript about the IDEAL measurement campaign. The IDEAL program and the associated campaign target a very relevant and interesting topic of atmospheric research, which is the structure of the lower troposphere in strongly stable conditions. New ways of sampling stability and turbulence with multiple UAS were explored in the campaign which could contribute significantly to a better understanding of the dynamics under such conditions.
The authors have revised the manuscript according to reviewer comments. It would have been easier to review the revised manuscript if a marked-up manuscript or a list of changes would have been provided as typically requested by AMT. It has not been clearly stated in the authors responses which changes were made, so a full review of the revised manuscript had to be made. Given the substantial changes that were made this was probably necessary in any case.
From my perspective, the manuscript has been improved significantly from the previous version. Nevertheless there are three major points which I think should be addressed before acceptance for publication in AMT and some minor comments:

Major points:
- Table 2: The authors should carefully revise the accuracy and resolution values that are given in this table and distinguish between laboratory or theoretical values and validated uncertainties in field experiments (incl. references).
- Methods: The authors refer to the doctoral thesis Doddi 2021 for many of the methods. The reference that is given is not traceable, no DOI, no book reference. It might be my fault that i cannot access the full text, but at this point I cannot judge if the presented results are fully traceable.
- Section 4: A series of plots are provided that show processed data from the UAS flights (Figs.10-19). Not all of the plots are discussed and it is questionable if they are relevant to the goals of the manuscript. In any case the authors should make clear, what the message of Section 4 is and clearly describe the findings with observations that are shown in the figures. This has improved and is maybe acceptable for Figs. 10-13, but especially for the new figures with regards to the spatial sampling, the description is very vague.

Minor comments:
- p.1, l.4f: I think that references should be omitted in the abstract

Noted. References are no longer cited in the abstract.

- p.1, l.16: "Atmospheric modeling motivated by IDEAL observations is reported elsewhere" That is a rather odd statement. Is it relevant in the abstract, if it is reported "elsewhere"?

This statement is omitted in the revised manuscript.

- p. 7, l. 154f: Although a time constant of 0.5 miliseconds and 800 Hz sampling rate might be technically correct for the coldwire sensors, how realistic are these characteristics? Is there a reference that shows that noise-free measurements are possible with a given resolution of 0.003 K in flight? Same applies to the hot wire. Spectra in Fig. 9 show significant noise from 100 Hz.

The time constant is a characteristic of the sensor wire at the mean airspeed and the electronics sampling rate is fixed. This determines the bandwidth that fluctuations can be measured, with roll off due to time constant at 320 Hz and Nyquist frequency at 400 Hz. Noise level is an independent issue,

affected by signal quantization, electronics noise, and vibration disturbances.

- p.8, Table 2: 0.01% is not a realistic noise-free resolution for the SHT sensor.

This is the value quoted on the manufacturer's spec sheet.

- p.13, l.239f: How did the changing atmospheric features result in aircraft sorties? Maybe rephrase?

The authors meant to state that the rapidly changing atmospheric conditions around the Granite mountain influenced the choice of number of aircraft deployed in each sortie.
This statement in its current phrasing is unclear and will be revised for clarity.

- p.13, l.240: In the author response it is written that wind speeds that exceed 20 m/s are frequently observed above 3000 m and are the limit for operation and here 15 m/s are given. Please just clarify what are the operational limits and why the 3000 m are set.

The mean wind speed bound for operation of the UAS is 15 m/s as stated in the manuscript. This allows for operation in the presence of gusts up to 20 m/s (the top airspeed in level flight). The mean wind speeds above 3000 m frequently exceeded 20 m/s during IDEAL campaign. Therefore, 3000 m was chosen as the ceiling for UAS operations, and operations near this limit were carefully supervised to prevent the aircraft from being blown downwind.

- p. 18, Table 3: I think the table is very helpful and informative, but maybe the full table could go to the appendix.

The authors recognize this concern and revise the manuscript to contain a few notable UAS sorties in the main document and migrate the full table to the appendix as recommended.

- p.19, l.265: It is uncommon to state whose doctoral thesis a reference is in the text. Just give the reference. However, the reference that is given links to a preview-website, which does not give the full text. I cannot access it.

The links embedded in this reference were found to be temporary. The authors were not aware of this problem. This reference will be revised to contain an accessible (full document) version of the doctoral thesis document.

- p.19, l.265: "was ?? to the DH2"
- p.19, l.266f: "accurate up to 0.05 m/s": Table 2 claims a resolution of 0.05 m/s and accuracy of 0.5 m/s. I think an accuracy of 0.05 m/s is quite unrealistic, considering the uncertainty of all the involved measurements.

The accuracy mentioned in Table 2 is correct. The value stated here i.e., 0.05 m/s is a typographical error and will be changed to 0.5 m/s in the revised manuscript.

- p.19, Fig. 9 and l.275ff: To my understanding, these are spectra of single time series with a length of 5 seconds. Averaged spectra would be interesting to see the characteristics of the sensor with less noise. Are these so-called artifacts systematic or random?

The raw spectrum computed using the 5s time intervals (solid blue lines in Figure 9) depict the instrument noise (at 100Hz in the top panel of Figure 9). This artifact is also captured in the bin averaged spectrum (red dots in Figure 9). This is due to the DH2 motor vibration and is systematic.

The artifacts due to DH2 motor vibrations are observed in the pitot data predominantly during ascent flight legs due to increased throttle setting; descent legs are typically free of these artifacts. Hotwire velocity is free of these vibrational artifacts in ascent as well as descent.

- p.24, l.288f: There is a verb missing in this sentence.

This will be corrected in the revised manuscript.

Figures 11 & 13: What do the thick and thin lines represent in the dissipation rate and Ct2 plots? It should be mentioned in the caption.

The thin lines represent the baseline offset for each ascent (red) or descent (blue) legs from the flight. The thick lines are the data. The figure captions will be updated to reflect this information in the revised manuscript.

- p.31, ll. 336ff: These statements should go into a "Data availability" section.

This paragraph will be moved to a new section describing Data access and availability.

**Reviewer 2: Comments**

Overview
The authors have answered most of my questions and objections satisfactorily. The structuring of the article is much improved, as is the quality of the figures. Useful clarifications have been added. The list of references has also been completed.

However, I still regret the absence of an evaluation of the impact of instrumental noise. Except for the position of the fitting line on the PSDs, no error bar is indicated, either on the measurements (T, velocities), or on the deduced quantities (theta, N2, Ri, epsilon, CT2). Taking into account the instrumental noise would certainly be a plus for such a paper presenting data and analysis methods. About the only error bar shown, on the fit of the -5/3 slope line on the PSDs of T and v, how is it estimated? (it is not an uncertainty on the slope of the line since it is fixed a priori).

Conclusion: in view of the improvements made to the manuscript, I consider that it can be published in AMT. I recommend, however, to include a consideration of instrumental noise, either by specifying the uncertainties on the measurements and the inferred quantities (could be done in the text), or by explaining why it is not possible to estimate them.

Reviewer 2 raises similar concerns to reviewer 1's Major Point #1 – regarding instrument noise and the role of sensor uncertainty in determining the accuracy of measured and estimated quantities. These issues have been addressed in the revised manuscript.